# Learning Node Selection via Tripartite Graph Representation in Mixed integer linear programming

## Abstract

Branch-and-bound methods are pivotal in solving Mixed Integer Linear Programs (MILPs), where the challenge of node selection arises, necessitating the prioritization of different regions of the space for subsequent exploration. While machine learning techniques have been proposed to address this, two crucial and open questions concerning **(P1)** the representation of the MILP solving process and **(P2)** the qualification of nodes in node selection remain open. To tackle these challenges, we propose a novel tripartite graph representation for the branch-and-bound search tree, which theoretically proves to effectively encapsulate the essential information of the search tree for node selection. Furthermore, we introduce three innovative metrics for node selection and formulate a Graph Neural Network (GNN) based model, named DQN-GNN, utilizing reinforcement learning to derive node selection policies. Empirical evaluations illustrate that DQN-GNN markedly enhances the efficiency of solving MILPs, surpassing the existing human-designed and learning-based models. Compared to other AI methods, our experiments substantiate that DQN-GNN exhibits commendable generalization to MILPs that are substantially larger than those encountered during training.

## 1 Introduction

Corporate decision-making paradigms are undergoing significant transformations, transitioning from traditional manual approaches to mathematical modeling and solver-based techniques (Zhang et al., 2023). This transformation is notably prevalent in numerous areas that encounter problems with integer variables, such as industrial process scheduling (Floudas & Lin, 2005), resources allocation (Ren & Gao, 2010), and logistic operations (Paschos, 2014). This has led to an increased focus on Mixed Integer Linear Programming (MILP), an essential type of mathematical problem that can effectively handle these integer constraints.

**Solving MILPs.** However, deploying MILP in real-world scenarios unveils pronounced intricacies. The inherent NP complexity (Paulus et al., 2022) and high dimensionality(Urbanucci, 2018) frequently push computational resources to their limits, especially when solutions are needed within stringent deadlines. Most MILP solvers (Gurobi., 2021; Bestuzheva et al., 2021; CPLEX., 2020) rely on the branch-and-bound (B&B) algorithm (Land & Doig, 2010), which recursively divides the search space into a tree. Through the decision tree, myriad pivotal decisions are repeatedly made (Linderoth & Savelsbergh, 1999), including determining the node to explore, selecting the branching variable, or even choosing the suitable heuristic for a given node (Fischetti & Lodi, 2010). These modules dictate the overall efficiency of identifying the optimal solution (Kianfar, 2010).

Unlike variable selection, which has a theoretically optimal strategy (strong branching) (Applegate et al., 1995), node selection lacks a universally acknowledged optimal method (He et al., 2014). Although currently neural network models are useful for solving MILP problems, designing an learning-based methods is still particularly challenging. It requires that neural networks have sufficient power to recognize key characteristics of MILPs and the search tree.

**Node Selection in B&B.** The overarching goal of the entire branch-and-bound algorithm is to accurately identify an integer solution and subsequently affirm its optimality (Boyd & Mattingley, 2007). To achieve this, a meticulous exploration of the entire feasible space is indispensable (Ibaraki, 1978).

Each node signifies a specific subspace, and the decision to expand a node involves an exploration into that subspace (Mitten, 1970). Upon the identification of an integer solution, the global lower bound (Norkin et al., 1998) is updated correspondingly, as the optimal solution should inherently be greater than or equal to the discovered solution. When exploring a node (a subproblem), if the Linear Programming (LP) relaxation solution of the node (its upper bound) surpasses the global lower bound, the node is deemed unfit for further exploration and is pruned (Yanover et al., 2006). That is because we can conclusively infer that a solution greater than the global lower bound doesn't reside in this node. This systematic approach gradually narrows the gap between the upper and lower bounds until a convergence to zero is attained (Huang et al., 2023). Efficient node selection is crucial in accelerating this convergence process (He et al., 2014; Yilmaz & Yorke-Smith, 2020; Labassi et al., 2022).

**Goals.** In this paper, we focus on the node selection process within the B&B algorithm, a critical yet relatively less explored decision task compared to other tasks, such as variable selection (Gasse et al., 2019; Gupta et al., 2020; Zarpellon et al., 2021; Nair et al., 2020; Etheve et al., 2020). This paper tries to address two fundamental but open problems for node selection in MILP: **(P1)** how to formulate a representation that accurately encapsulates both the inherent properties of MILP and the insights obtained during the solving process to select appropriate nodes, and **(P2)** how to measure the "goodness" of a node.

**Analysis for P1.** Determining what constitutes as "sufficient" information is pivotal for inferring the optimal node for selection, a task often overshadowed by the prevailing approach that perceives each newly expanded node as an isolated subproblem. We advocate for a perspective that perceives each selected node as a nuanced divergence from its parent, distinguished by a newly added constraint. This approach emphasizes the subtle distinctions between proximate nodes and minimizes redundancy in representing the foundational problem. This innovative viewpoint strives to offer a holistic representation by amalgamating inherent node information and insights acquired subsequent to node selection. This accentuates the intrinsic interconnectedness of the subproblem with the overarching problem. A more detailed exploration of this perspective is delineated in Section 3.

**Analysis for P2.** To address this point, we reexamine the primary objective of node selection: minimizing the overall solving time (He et al., 2014). The solver concludes its process once the global gap reaches zero, thus accelerating gap convergence is our primary objective. However, solely relying on the gap for training is insufficient as the gap only changes when a new feasible solution is discovered (Mahmoud & Chinneck, 2013). To address this, we incorporate a second guiding principle: leveraging the path to the historical optimal solution, a strategy proven effective in previous works (He et al., 2014; Yilmaz & Yorke-Smith, 2020; Labassi et al., 2022). Additionally, the time spent in the node selection process is also a significant factor to consider. A crucial, yet often overlooked aspect in node selection is the transition from the current focus node to the newly selected one. We delve into these questions in detail in Section 4.

**Contributions.** Our main contributions are summarized as follows:

• **Novel Representation.** We have introduced a novel tripartite graph representation for the branch-and-bound search tree, addressing the significant issue of inadequate representation of the MILP intermediate solving process. This representation is theoretically proven to encapsulate sufficient information of the search tree, enabling effective node selection in the solving process.

• **Metrics and Model Development.** We have proposed three innovative metrics for node selection and developed a DQN-GNN model. This model employs reinforcement learning to learn node selection policies in MILP solving.

• **Empirical Validation.** We design and conduct experiments that demonstrate the efficacy of the DQN-GNN model in enhancing the efficiency of solving MILPs. The model has demonstrated significant improvement over existing human-designed and learning-based benchmarks.

## 2 PRELIMINARIES

In this section, we present concepts and definitions that will be used throughout this paper. We first describe how to represent an MILP with a weighted bipartite graph, then we define the branch-and-bound process with strict mathematical definitions.

**MILP as Weighted Bipartite Graph.** A general MILP problem is defined by a set of decision variables, where a subset or all variables are required to be integers. The objective is to maximize a linear function under a series of linear constraints, as formulated below:

$$
\begin{aligned}
\max \quad & \boldsymbol{c}^\top \boldsymbol{x} \\
\text{s.t.} \quad & \boldsymbol{A}\boldsymbol{x} \geq \boldsymbol{b}, \boldsymbol{x} \in \mathbb{R}^N, \\
& x_j \in \mathbb{Z}, \forall j \in \mathcal{I},
\end{aligned}
\tag{1}
$$

For simplicity, we assume that the objective of the MILP problems discussed in this paper is to seek the maximum value.

Prior to developing graph representations for MILPs, we commence by defining the specific type of graph to be employed subsequently: a weighted bipartite graph. The weighted bipartite graph $G = (V \cup C, E)$ consists of a vertex set $V \cup C$ that are divided into two groups variable vertex set $V$ and constraint vertex set $C$ with $V \cap C = \emptyset$, and a collection $E$ of weighted edges, where each edge connects exactly one vertex in $V$ and one vertex in $C$. There is an edge between a variable vertex and a constraint vertex if the variable has a nonzero coefficient in the constraint. Note that there is no edge connecting vertices in the same vertex group. $E$ can also be viewed as a function $E : V \times C \to \mathbb{R}^e$, where $e$ denotes the dimension of the edge attributes. We use $\mathcal{G}_{l,m}$ to denote the collection of all weighted bipartite graphs $G = (V \cup C, E)$ with $|V| = l$ and $|C| = m$. We always write $V = \{v_1, v_2, \ldots, v_l\}$, $C = \{c_1, c_2, \ldots, c_m\}$, and $E_{i,j} = E(v_i, c_j)$, for $i \in \{1, 2, \ldots, l\}, j \in \{1, 2, \ldots, m\}$.

One can equip each vertex and edge with a feature vector. Throughout this paper, we denote $h_i^V \in \mathcal{H}^V$ as the feature vector of vertex $v_i \in V$, $h_j^C \in \mathcal{H}^C$ as the feature vector of vertex $c_j \in C$, and $h_k^E \in \mathcal{H}^E$ as the feature vector of edge $e_k \in E$, where $\mathcal{H}^V, \mathcal{H}^C, \mathcal{H}^E$ are feature spaces. Then we define $\mathcal{H}^V{}_l := (\mathcal{H}^V)^l$, $\mathcal{H}^C{}_m := (\mathcal{H}^C)^m$ and concatenate all the vertex features together as $H^V = (h_1^V, h_2^V, \ldots, h_l^V) \in \mathcal{H}_l^V$, $H^C = (h_1^C, h_2^C, \ldots, h_m^C) \in \mathcal{H}_m^C$. Finally, a weighted bipartite graph with vertex features is defined as a tuple $(G, H^V, H^C) \in \mathcal{G}_{l,m} \times \mathcal{H}_l^V \times \mathcal{H}_m^C$.

With the concepts described above, one can represent an MILP as a bipartite graph (Nair et al., 2020; Gasse et al., 2019): Each vertex in $V$ represents a variable in MILP and each vertex in $C$ represents a constraint.

**Branch-and-Bound Algorithm.** The B&B algorithm, a well-regarded tree search method, is commonly used to solve MILP problems. The strategy behind B&B involves a divide-and-conquer approach, breaking down the search space by branching on variable values. During the solving process, a search tree $T$ is constructed. Each node in the tree corresponds to a subproblem of the original MILP but with additional constraints. The nodes which we have not explored are called open nodes. Our attention is firmly directed towards the **node selection policy** $\pi_{ns} : \mathcal{N} \times \mathcal{S} \to ns \in \mathcal{N}$, which guides the choice of a node $ns$ from the open node set $\mathcal{N}$ according to the search tree state space $\mathcal{S}$. Then, the LP relaxation of this node can be solved, where all variables are treated as continuous. Efficient algorithms such as the simplex method can be utilized to solve this equation, and the optimal solution $x_{ns}$ thus obtained provides a lower bound $f(x_{ns})$ for the original MILP problem.

If the LP relaxation solution $x_{ns}$ of the selected node violates the original integrality constraints, the problem "branches" into two new subproblems (child nodes) by adding constraints that compel the fractional variable to round up or down. Specifically, the leaf node is added with constraints $x_i \leq \lfloor (x_{ns})_i \rfloor$ and $x_i \geq \lceil (x_{ns})_i \rceil$, respectively, where $x_i$ denotes the $i$-th variable, $(x_{ns})_i$ denotes the $i$-th entry of vector $x_{ns}$, and $\lfloor \rfloor$ and $\lceil \rceil$ denote the floor and ceil functions. In contrast, if the solution $x_{ns}$ is integer (and feasible for the original MILP as per Equation 1), and its objective value surpasses the current best integer feasible solution, it is designated as the new global lower bound. Alternatively, if the objective value $f(x_{ns})$ (i.e., the node upper bound) is lower than the global lower bound, or if the LP problem is infeasible, the node is pruned.

## 3 Proposed Representation of Branch-and-Bound Tree

In this section, we initially elucidate our motivation and define the structure of the tripartite graph, subsequently demonstrating our findings on why the information encapsulated within the tripartite graph suffices for effective node selection.

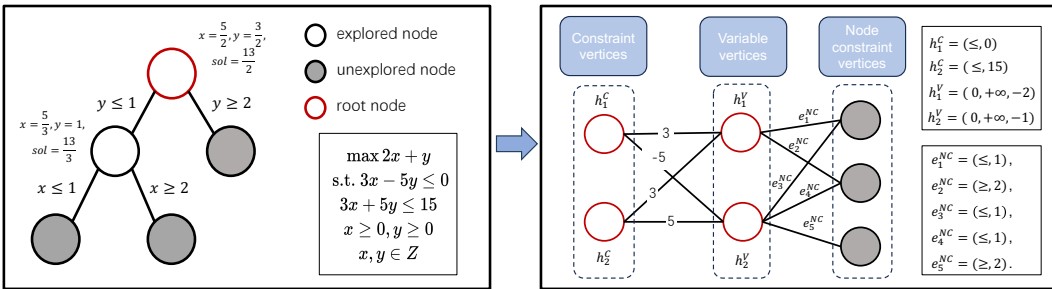

Figure 1: Example of tripartite graph representation. The root node (red) is conceptualized as a bipartite graph, consisting of variable and constraint vertices, while the leaf nodes (grey) embody sets of newly incorporated constraints. The features of the edges, which connect the variable vertices to the node constraint vertices, delineate the constraint space of the leaf nodes.

## 3.1 MOTIVATION

In addressing the critical question (**P1**), we focus on how to refine the representation of newly explored information in MILP problems, particularly within the branch-and-bound (B&B) process. An essential aspect of this exploration is determining what information to preserve and represent at each stage of the search tree.

**Node Information for Selection.** To provide a more nuanced measure of a node's exploration potential, we introduce the concept of node information, denoted as $I(N)$. This concept captures all relevant knowledge gained following the selection of node $N$, which is instrumental in steering the search process effectively. It consists of two principal components: **(1)** Inherent Information, $I_1(N)$: This component comprises the set of constraints that are explicitly associated with node $N$, which are the direct result of the specific branching decisions taken to reach $N$ from the root of the search tree. Immediately upon selection, $I_1(N)$ is available, which delineates the search path of $N$ and indicates its intrinsic potential within the search space. **(2)** Derived Information, $I_2(N)$: Following node selection, $I_2(N)$ emerges from solver computations and heuristic algorithms, offering dynamic updates to the bounds.

**The Limitations of the Bipartite Graph Model.** Previous studies, notably by Labassi et al. (2022), have adopted a bipartite graph model to represent nodes in the B&B tree, treating each as an independent subproblem with its own set of variables and constraints. While this approach offers a comprehensive view of each node, especially in terms of $I_1(N)$, it also introduces significant informational redundancy. This is because many nodes share a large portion of their constraints and variables. Such redundancy can obscure the subtle, yet critical, differences between closely related nodes that arise from distinct branching decisions.

**From Subproblems to Constraints.** We observe that subproblems in MILP do not arise in isolation; they are developed step-by-step through a series of branching decisions. Each branching operation introduces new constraints, subtly altering the problem space. These incremental changes are critical for AI to learn the distinctions between nodes, which is often lost in traditional representations. To address these limitations, we propose a tripartite graph model. This model maintains the root node's complete bipartite graph representation, capturing the original problem's full scope. For subsequent nodes, however, we shift our focus to the constraints added through branching.

## 3.2 REPRESENTING B&B TREES WITH TRIPARTITE GRAPHS FOR NODE SELECTION

Although we have seen each node in the tree as a set of constraints, we have left two issues unaddressed: whether these nodes are equally important and whether there is a need to represent all of them explicitly. We respond to these questions by introducing two theorems. The first theorem proves that the information of a node can be encompassed by the information of its expanded child nodes, which also provides an answer to the second question. Subsequently, we present the second theorem, asserting that the information from the root problem and the leaf nodes is indeed sufficient for node selection. The full proof is presented in the Appendix B.1.

**Theorem 3.1.** *Given a node $N_0$ and its two child nodes $N_1$ and $N_2$, it holds that $I(N_0) \subseteq I(N_1) \cup I(N_2)$, where $I(N)$ denotes the information of the node $N$.*

This theorem demonstrates that the information of a node can be encompassed by the information of its expanded child nodes. With Theorem 3.1 served as a foundation, we can prove that the information from the root problem and the leaf nodes is indeed sufficient for node selection.

**Theorem 3.2.** *Given a B&B search tree $T$, the entirety of its information pertinent to node selection can be encapsulated by the constraint and variable information of its root node, coupled with the constraint, upper bound, and lower bound information of the leaf nodes within the tree.*

The information encompassed within the search tree is twofold: it includes the original problem, represented by the root node, and the information from the explored nodes. As established in Theorem 3.1, the information contained within a parent node can be derived from the information within its child nodes. Consequently, for a given search tree, all explored information can be represented exclusively by the collection of its leaf nodes.

Then an MILP solving search tree can be represented as a tripartite graph. The root node, which embodies the original problem, is formulated as a bipartite graph, following the approach in Labassi et al. (2022), consisting of vertices for variables and constraints. In the search tree, each leaf node is a subproblem created through branching, where each branch adds a new constraint. To represent this in our tripartite graph, we use 'node constraint vertices' to represent the leaf nodes. A series of edges connect these vertices to 'variable vertices,' collectively representing the sequence of constraints that have been added throughout the branching process. We present the search tree in an MILP instance solving process and its corresponding tripartite graph in Figure 1.

### 3.3 TRIPARTITE GRAPH REPRESENTATION

Building upon the existing bipartite graph representation of the MILP root node problem, we extend this representation to encapsulate not only the inherent problem structure but also the intermediate exploration process during the solution. The node constraint is articulated as a set of constraints added to the root problem: $\{x_i \leq z_i | i \in \mathcal{I}, z_i \in \mathbb{Z}\}$ or $\{x_i \geq z_i | i \in \mathcal{I}, z_i \in \mathbb{Z}\}$. We integrate these new node constraint vertices into the original bipartite graph, observing that these vertices exclusively form edges with the variable vertices. As a result, we formulate a tripartite graph $G = (V \cup C \cup NC, E^C \cup E^{NC})$ which includes a vertex set $V \cup C \cup NC$, divided into three subsets: the variable vertex set $V$, the constraint vertex set $C$ and the node constraint vertex set $NC$, with $V \cap C = V \cap NC = C \cap NC = \emptyset$. It also encompasses a collection $E^C$ of weighted edges, each connecting vertices from $V$ and $C$, and $E^{NC}$ connecting vertices from $V$ and $NC$. We denote the collection of all such weighted tripartite graphs $G = (V \cup C \cup NC, E^C \cup E^{NC})$ with $|V| = l, |C| = m$ and $|NC| = n$ as $\mathcal{G}_{l,m,n}$. Further, we denote $V = \{v_1, v_2, \ldots, v_l\}, C = \{c_1, c_2, \ldots, c_m\}, NC = \{nc_1, nc_2, \ldots, nc_n\}$. The edges are denoted as $E_{i,j}^C = E^C(v_i, c_j)$ and $E_{j,k}^{NC} = E^{NC}(c_j, nc_k)$, with $|E^C| = e_1, |E^{NC}| = e_2$, for $i \in \{1, 2, \ldots, l\}, j \in \{1, 2, \ldots, m\}, k \in \{1, 2, \ldots, n\}$.

Each vertex and edge are associated with a feature vector. Let $h_i^V \in \mathcal{H}^V, h_j^C \in \mathcal{H}^C$, and $h_k^{NC} \in \mathcal{H}^{NC}$ represent the feature vectors of vertex $v_i \in V, c_j \in C$ and $nc_k \in NC$, respectively. Subsequently, we define $\mathcal{H}_l^V := (\mathcal{H}^V)^l, \mathcal{H}_m^C := (\mathcal{H}^C)^m$, and $\mathcal{H}_n^{NC} := (\mathcal{H}^{NC})^n$. We concatenate all the vertex features to form $H = (h_1^V, h_2^V, \ldots, h_l^V, h_1^C, h_2^C, \ldots, h_m^C, h_1^{NC}, h_2^{NC}, \ldots, h_n^{NC}) \in \mathcal{H}_l^V \times \mathcal{H}_m^C \times \mathcal{H}_n^{NC}$. The edge features are denoted as $f_i^{E^C} \in \mathcal{F}^{E^C}$ and $f_j^{E^{NC}} \in \mathcal{F}^{E^{NC}}$. We then define $\mathcal{F}_{e_1}^C := (\mathcal{F}^C)^{e_1}$ and $\mathcal{F}_{e_2}^{NC} := (\mathcal{F}^{NC})^{e_2}$ and concatenate them to obtain $F = (f_1^{E^C}, f_2^{E^C}, \ldots, f_{e_1}^{E^C}, f_1^{E^{NC}}, f_2^{E^{NC}}, \ldots, f_{e_2}^{E^{NC}})$.

## 4 LEARNING NODE SELECTION VIA GNN

Node selection in the B&B tree involves a series of sequential decisions, where each choice impacts the subsequent ones and the final result. Furthermore, the branch-and-bound process often encounters delayed rewards, meaning an early decision's consequences might only become apparent after several steps. Therefore, we leverage Reinforcement Learning (RL) to learn node selection policies. In this section, we initially address the question **(P2)**: How to quantify the "goodness"

of a node. Subsequently, we formulate our node selection problem as a Markov Decision Process (MDP). Finally, we provide an exhaustive description of our proposed DQN-GNN model.

## 4.1 QUANTIFYING NODE GOODNESS

Before delving into the intricacies of our reinforcement learning methodology, it is pivotal to address the crucial question (**P2**): How does one quantify the "goodness" of a node?

**Objective 1: Acceleration of Gap Convergence.** To address this, let's revisit the primary objective of node selection, which is to minimize the overall solving time. The solver concludes its process once the global gap reaches zero, making the acceleration of gap convergence our first objective.

**Sparsity of Gap.** However, relying solely on the gap for training proves to be insufficient. This is because the gap only undergoes changes when a new feasible solution is discovered, leading to updates in the lower bound, or when all nodes of identical depth have been explored, causing updates in the upper bound. Intuitively, the gap remains constant through the majority of the selection steps. The subsequent theorem elucidates the sparsity of the gap encountered during node selection. The detailed proof is relegated to Appendix B.2 due to space constraints.

**Theorem 4.1.** *Consider a B&B tree $T$ containing $s_0$ nodes. Suppose that, at each round $t$, the gap reward $r(\cdot)$ for the selected node $a_t$ is finite, denoted as $|r(\cdot)| \leq r_0$. Given an initial point $x'$ and a heuristic algorithm with exploration ability $\delta$, the algorithm can explore integer solutions in the space $\{x \in \mathbb{Z}^n \mid x'_i - \frac{\delta}{2} \leq x_i \leq x'_i + \frac{\delta}{2}, i = 1, 2, \ldots, n\}$. If $\delta \leq \sqrt[n]{\frac{\epsilon - (\log_2 s_0 + 1)/s_0}{r_0 \cdot s_0}}$, then it holds that $\mathbb{E}\left[\sum_t r(a_t)\right] \leq \epsilon$.*

**Objective 2: Historical Optimal Solution Path.** The sparsity of the gap complicates the learning process for the reinforcement model due to the lack of effective learning signals. To mitigate this, we employ a second guiding principle: leveraging the path to the historical optimal solution, a proven strategy in previous works (He et al., 2014; Yilmaz & Yorke-Smith, 2020; Labassi et al., 2022). Given that each node represents a specific search space, we can determine whether this optimal solution resides within the space of a particular node, indicating the potential of the node to lead to the optimal solution. This method facilitates quicker discovery of feasible solutions by utilizing historical information from similar problems.

However, we refine this approach by reducing the reward value in the early exploration stages. This adjustment is based on the rationale that in the initial steps, due to the vastness of the search space and the limited available information, establishing a connection between the information and the potential to reach the optimal solution is intricate. Moreover, discerning whether a node can lead to the optimal solution is not pivotal in the initial stages. The likelihood of the optimal solution residing within the explored space is high, and the predictions, being based on incomplete feature exploration, are not precise. Even if the initial selections deviate from the optimal solution path, it is not detrimental as our objective is not solely to find the optimal solution but also to ascertain the absence of superior solutions in other spaces. Thus, minor inaccuracies in the exploration of initial nodes do not critically impact the overall search and verification process.

**Objective 3: Path Switching Cost.** Our prior discussions have centered around strategies for accelerating gap convergence, but it's crucial to remember that our ultimate goal is to reduce the overall solving time. Thus, the time spent in the node selection process itself is also a substantial factor. A crucial, unaddressed question in node selection is how we transition from the current focus node to the newly selected one. In the branch-and-bound process, solvers (Bestuzheva et al., 2021) navigate the path from the focus node to the newly chosen one (path switch). As illustrated in Figure 2, both the focus and the newly chosen nodes repeatedly trace back to their parent nodes until a common ancestor node is found. In certain problems such as the Weighted

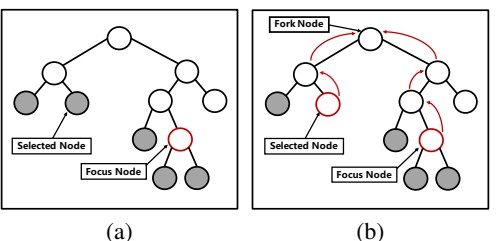

Figure 2: Path switching process.

Partial MaxSAT (WPMS) dataset with $n \in [70, 80]$, path switch phase averagely consumes 5.2% of the total solving time.

**Reward Function.** We formulate the reward function to encompass the three objectives discussed previously, structured in three components: **(1) Gap update reward.** If the gap updates, a fixed reward, $r_{gap}$, is received. If there is no update in the gap, the reward for this component is zero. **(2) Optimal solution path reward.** If the optimal solution resides within the current node's domain, a reward, $F$, is assigned. The value of $F$ is designed to be smaller in the initial steps but increases as it reaches deeper nodes. If the optimal solution is not within the domain of the current node, the reward for this component is zero. **(3) Path switching penalty.** We penalize path switching by subtracting a term proportional to the number of backtracking steps from the total reward.

**Node Selection's Relationship with Variable Selection.** A crucial point of focus is the interplay between variable selection and node selection. The outcome of variable selection is perceived to influence node selection and it cannot be exclusively modulated by the node selector. However, we introduce a novel perspective, asserting that while the variable selection process shapes the tree, the node selection strategy can independently assess the tree's current state to determine the next course of action. Once a specific subtree is formed, the node selection process focuses solely on this existing state of the tree to identify the most promising node for exploration. This decision-making process within the node selection phase does not require direct knowledge of the variable selection policies that led to the current state of the tree, as articulated in Theorem 4.2.

**Theorem 4.2.** *Consider a node selection policy, denoted as $\pi_{ns}$, that makes selections based on the three objectives defined in Section 4. For any given variable selection policy $\pi_{vs}$, we posit that the optimal node choice, denoted as $ns$, does not rely on the specifics of $\pi_{vs}$ for its determination.*

The detailed proof is relegated to Appendix B.3 due to space constraints. Consequently, the node selection strategy itself does not need to factor in the specifics of the variable selection decisions. Instead, it can effectively operate based on the current state of the tree, regardless of the variable selection path that led to it.

## 4.2 REINFORCEMENT LEARNING FORMULATION

**Markov Decesion Process (MDP).** We formulate an MILP solver as the environment and the RL model as the agent. We consider an MDP defined by the tuple $(\mathcal{S}, \mathcal{A}, r, \pi)$. Specifically, we specify the state space $\mathcal{S}$, the action space $\mathcal{A}$, the reward function $r : \mathcal{S} \times \mathcal{A} \to \mathbb{R}$, the transition function $\pi$, and the terminal state in the following. **(1) The state space, $\mathcal{S}$:** As delineated in Section 3, the core information for node selection is represented by a tripartite graph. **(2) The action space $\mathcal{A}$.** The action space is intended to include all nodes that are potentially selectable. However, the dynamic nature of the selection process means the number of selectable nodes is subject to change due to the addition of newly expanded nodes and the removal of pruned nodes. To address this variability, we employ the heuristic node selection algorithm called $Estimate$ in modern solvers to pre-select nodes, choosing the top $n$ nodes, where $n$ is a predetermined value, to form a set of node candidates. If the initial set of candidates is less than $n$, placeholders are used to fill the remaining slots, ensuring a consistent set size. We define the action space as this set of node candidates with a size of $n$. **(3) The reward function $r$.** The reward function, as previously discussed, encompasses the gap update reward, the optimal solution path reward, and the path switching penalty. **(4) The transition function $\pi$.** The transition function maps the current state $s$ and the action $a$ to the next state $s'$, representing the ensuing search tree post the expansion of node $a$. **(5) The terminal state.** The process reaches a terminal state when the gap attains zero or there are no remaining candidate nodes in the set.

This MDP framework diverges from previous studies, which predominantly emphasized selecting leaf nodes, akin to a Depth-First Search (DFS) strategy, where the agent typically chooses between two child nodes. In contrast, our method contemplates a broader range of potential nodes. Within this framework, each episode is equivalent to solving an MILP instance, with initial states representing instances sampled from a specific group. The trajectory probability $\tau = (s_0, \ldots, s_T)$ is contingent upon both the node selection policy $\pi$ and the other solver components, formulated as $p_\pi(\tau) = p(s_0) \prod_{t=0}^{T-1} \sum_{a \in \mathcal{A}} \pi(a|s_t) p(s_{t+1}|s_t, a)$.

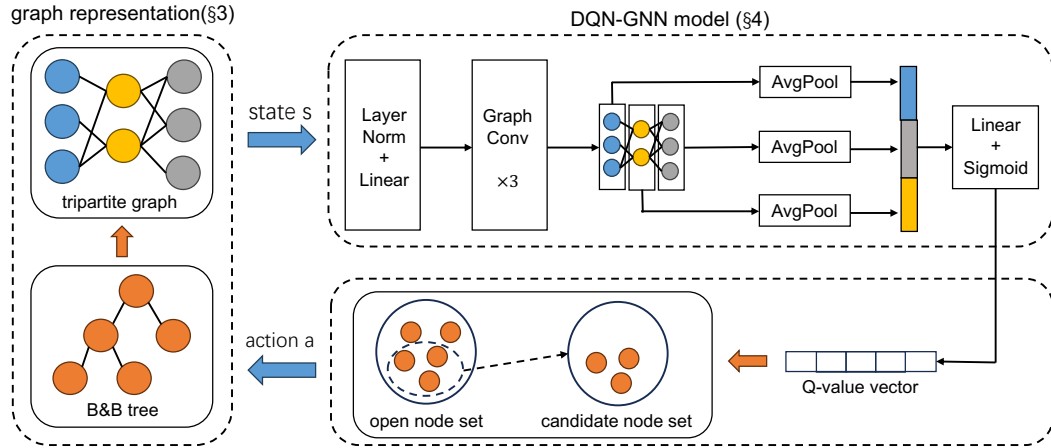

Figure 3: Illustration of our proposed RL framework for learning node selection policies. In this framework, the search tree is represented as a tripartite graph, serving as the environment, and the DQN-GNN model acts as the agent.

**DQN-GNN Model.** Reinforcement learning is designed to learn an approximately optimal policy: a function that maps states to actions, such that the accumulated reward is maximized (Sutton & Barto, 2018). Figure 3 delineates the architecture of our proposed model. Our model is developed based on the foundational work of Labassi et al. (2022). Additional details on the model architecture are included in Appendix D. A significant advantage of this model is its ability to accommodate MILPs with varying numbers of variables and constraints. Moreover, the model is adept at adapting to the dynamic nature of the branch-and-bound tree, where the number of nodes is subject to change during the solving process. This adaptability is facilitated by converting the search tree information into a standardized graph format, allowing for the consistent training of the model regardless of the dynamic variability in the tree's structure.

## 5 EXPERIMENTS

Our experiments have two main parts: **Experiment (1)** Evaluate our approach on three classical MILP problems. **Experiment (2)** Test whether DQN-GNN can generalize to instances significantlly larger than those seen during training. The codes are modified from Labassi et al. (2022).

**Benchmarks.** We evaluate our approach on three NP-hard MILP problem benchmarks, including Fixed Charge Multicommodity Network Flow (FCMCNF) (Hewitt et al., 2010) , Weighted Partial MaxSAT (WPMS) (Ansótegui & Gabàs, 2017) and Generalized Independent Set (GISP) instances (Colombi et al., 2017). We artificially generate instances following Béjar et al. (Béjar et al., 2009); Chmiela et al. (Chmiela et al., 2021). Due to limited space, please see Appendix E.1 for details of these datasets.

**Baselines.** We compare against the state-of-the-art best estimate node selection rule (Bénichou et al., 1971; Forrest et al., 1974). This is the default method in SCIP (Bestuzheva et al., 2021). Besides, we also report the results with a plain rule that always selects the highest-ranked node (ESTIMATE). In addition, we compare against three machine learning approaches: the Support Vector Machine (SVM) approach (He et al., 2014), the RankNet feedforward neural network approach (Song et al., 2018), and the approach based on Graph Neural Networks (GNN) (Labassi et al., 2022).

**Experimental Setup.** Throughout all experiments, we use SCIP 8.0.4 (Bestuzheva et al., 2021) as the backend solver, which is the state-of-the art open source solver, and is widely used in research of machine learning for combinatorial optimization (Chmiela et al., 2021; Gasse et al., 2019; Turner et al., 2022; Wang et al., 2023). Additional details on the experimental setup and hardware specification are included in Appendix E.3.

**Evaluation Metrics.** We employ two widely recognized evaluation metrics: solving time (Time, lower is better), and the branch-and-bound tree size (Nodes, lower is better). It is crucial to underscore that in the context of solver processes, time is the paramount metric, as the temporal cost is

highly valuable, whereas the space occupied by the search tree nodes is relatively limited. We introduce the Nodes metric primarily as a supplementary measure to provide additional insights. We assess node selection methods in terms of the 1-shifted geometric mean over the instances, accompanied by the geometric standard deviation.

**Experiment (1): Comparative Evaluation.** For each problem, machine learning models are trained on instances of the same size as the test instances (50 instances). The results in Table 1 suggest DQN-GNN significantly outperforms all the baselines on three MILP problems. Compared to SCIP, DQN-GNN demonstrates notable efficiency improvements across all tested problems, being approximately 17.24% faster in FCMCNF, 17.17% in WPMS, and 3.74% in GISP.

Table 1: Comparison of Average Solving Time and B&B Tree Size (Test).
FCMCNF: $n = 15$ nodes. WPMS and GISP: number of nodes $n \in [60, 70]$.

| Methods | FCMCNF | | WPMS | | GISP | |
|---|---|---|---|---|---|---|
| | Time(s) | Nodes | Time(s) | Nodes | Time(s) | Nodes |
| SCIP | 4.64 ± 1.38 | 28.11 ± 4.56 | 12.59 ± 1.70 | 199.97 ± 2.00 | 3.74 ± 1.34 | 84.05 ± 3.69 |
| ESTIMATE | 4.17 ± 1.36 | 38.40 ± 3.71 | 10.59 ± 1.54 | **199.90 ± 1.69** | 3.73 ± 1.38 | 79.03 ± 3.32 |
| SVM | 4.18 ± 1.40 | 33.44 ± 3.63 | 12.03 ± 2.09 | 250.07 ± 3.17 | 3.73 ± 1.35 | 89.33 ± 3.46 |
| RankNet | 3.92 ± 1.37 | **21.55 ± 3.39** | 11.52 ± 1.93 | 212.09 ± 2.71 | 3.77 ± 1.37 | 91.52 ± 3.25 |
| GNN | 4.03 ± 1.38 | 24.71 ± 3.42 | 12.07 ± 2.10 | 215.22 ± 2.76 | 3.79 ± 1.36 | 93.36 ± 3.26 |
| DQN-GNN (Ours) | **3.84 ± 1.33** | 29.37 ± 4.48 | **10.41 ± 1.95** | 204.70 ± 2.79 | **3.60 ± 1.34** | **78.31 ± 3.48** |

**Experiment (2): Generalization.** We evaluate the ability of DQN-GNN to generalize across larger sizes of MILPs. We evaluate these models on the larger transfer instances (50 instances). From Table 2, several key observations can be made. Firstly, SCIP, as a conventional method, demonstrates superior performance in most cases, especially in terms of time and the number of nodes. This indicates that although AI methods possess potential and extensibility in solving MILPs, they still fall short in some aspects compared to classic methods like SCIP. However, it is pivotal to explicitly state that among AI methods, DQN-GNN has already achieved the best extensibility compared to other AI methods. In terms of average solving time, DQN-GNN is optimal on two of the datasets, surpassing all other AI-based approaches. Specifically, on the WPMS dataset, DQN-GNN even excels beyond SCIP, indicating its superiority in certain specific scenarios.

Table 2: Comparison of Average Solving Time and B&B Tree Size (Transfer).
FCMCNF: $n = 20$ nodes. WPMS and GISP: number of nodes $n \in [70, 80]$.

| Methods | FCMCNF | | WPMS | | GISP | |
|---|---|---|---|---|---|---|
| | Time(s) | Nodes | Time(s) | Nodes | Time(s) | Nodes |
| SCIP | **10.70 ± 2.12** | **29.72 ± 7.28** | 24.62 ± 1.76 | **413.54 ± 1.24** | **5.99 ± 1.24** | **193.61 ± 1.83** |
| ESTIMATE | 16.25 ± 2.17 | 123.45 ± 10.75 | 21.87 ± 1.64 | 466.09 ± 1.51 | 6.27 ± 1.45 | 370.11 ± 2.08 |
| SVM | 14.47 ± 2.02 | 50.92 ± 7.66 | 20.36 ± 1.71 | 563.75 ± 2.04 | 8.36 ± 1.25 | 447.73 ± 1.64 |
| RankNet | 12.85 ± 1.84 | 32.22 ± 5.61 | 25.12 ± 1.70 | 654.55 ± 1.63 | 8.61 ± 1.25 | 415.27 ± 1.73 |
| GNN | 13.57 ± 1.77 | 44.14 ± 5.720 | 28.39 ± 1.55 | 841.35 ± 1.63 | 8.11 ± 1.29 | 342.94 ± 2.13 |
| DQN-GNN (Ours) | 13.55 ± 2.01 | 43.54 ± 6.03 | **19.01 ± 2.12** | 652.32 ± 1.82 | 7.16 ± 1.20 | 308.45 ± 2.24 |

# 6 CONCLUSIONS

We addressed two pivotal and open questions concerning the inadequate representation of the MILP intermediate solving process and the qualification of nodes during node selection. We introduce an innovative tripartite graph representation for the branch-and-bound search tree and provide theoretical evidence demonstrating that our tripartite graph can adequately represent the information of the search tree required for effective node selection. Subsequently, we introduce three metrics for node selection and develop a novel DQN-GNN model, leveraging reinforcement learning to acquire node selection policies. Experimental results reveal that the DQN-GNN model markedly enhances the efficiency of solving MILPs, outperforming both human-designed and other learning-based benchmarks. We are confident that our proposed methodology offers fresh perspectives and insights into learning node selection strategies, paving the way for further advancements in this domain.

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

# A  RELATED WORK

**Machine Learning for MILP.** Real-world MILP problems often maintain consistent structural patterns, even with minor alterations in constraints or objectives. Traditionally, these would be made according to hard-coded expert heuristics (Achterberg, 2007) implemented in solvers (Bestuzheva et al., 2021; CPLEX., 2020; Gurobi., 2021). Recently, however, machine learning methods have emerged as potent solutions for MILP problems (Bengio et al., 2021). Several machine learning approaches have been explored for solving MILP problems within branch-and-bound frameworks (Gasse et al., 2019; Lodi & Zarpellon, 2017; He et al., 2014). Leveraging the knowledge from prior instances, AI adapts efficiently to new yet structurally similar problems. These AI-driven methods don't operate independently of the B&B algorithm. Instead, they work in synergy, primarily by replacing manually crafted expert strategies, which ensures that decisions within the B&B modules are more attuned to the problem's unique structure. This line of research has shown significant improvement on the solver performance, including node selection (He et al., 2014; Song et al., 2018; Labassi et al., 2022; Yilmaz & Yorke-Smith, 2020), cut selection (Tang et al., 2020; Wang et al., 2023; Paulus et al., 2022; Turner et al., 2022; Baltean-Lugojan et al., 2019), variable selection (Khalil et al., 2016; Gasse et al., 2019; Gupta et al., 2020; Balcan et al., 2018; Zarpellon et al., 2021), column generation (Morabit et al., 2021), and heuristics (Khalil et al., 2017; Hendel et al., 2019; Shen et al., 2021). In this paper, we focus on node selection, which plays a significant role in modern MILP solvers (He et al., 2014).

For node selection, He et al. (He et al., 2014) took the first strides towards leveraging machine learning for node comparison heuristics in branch-and-bound. They proposed training a support vector machine (SVM) to emulate the diving oracle's node comparison operator. Their work, however, was limited to conjunction with a learned pruning model, yielding a method more akin to a primal heuristic. Subsequently, Song et al. (Song et al., 2018) trained a multilayer perceptron RankNet model for node comparison, introducing retrospective imitation learning. They achieved significant improvements on specific path planning integer programs but reported less impactful results on more complex benchmarks. Yilmaz and Yorke-Smith (Yilmaz & Yorke-Smith, 2020) proposed a limited form of feedforward neural network node comparison operator, combined with a backtracking algorithm, to provide a full node selection policy. They reported improvements in time and the number of nodes on several benchmarks.

**GNN in MILP.** Graph Neural Networks (GNNs) are specialized neural networks designed for graph-structured data and have found extensive applications across diverse domains such as recommender systems, traffic analysis, and chemistry, to name a few (Wu et al., 2020; Zhou et al., 2020). The integration of GNNs to expedite optimization solvers has garnered increasing attention recently (Peng et al., 2021; Cappart et al., 2023). Numerous graph-related optimization problems, including minimum vertex cover, traveling salesman, and vehicle routing, can be aptly represented and approximately solved using GNNs, owing to their inherent problem structures (Khalil et al., 2017; Kool et al., 2018; Joshi et al., 2019; Drori et al., 2020). Furthermore, GNNs can be instrumental in solving general Mixed-Integer Linear Programming (MILP). Gasse et al. (2019) pioneered the representation of an MILP using a bipartite graph and leveraged a GNN to guide an MILP solver. Following this, a myriad of approaches have emerged, utilizing GNNs to guide MILP or LP solvers (Ding et al., 2020; Nair et al., 2020; Gupta et al., 2020; Paulus et al., 2022; Khalil et al., 2022; Shen et al., 2021; Labassi et al., 2022). Despite the promising empirical outcomes observed, there remains a discernible gap in the design of representations for the MILP solving process.

# B  THEORETICAL ANALYSIS

We enumerate our theoretical findings and provide detailed proofs to substantiate them.

## B.1  THEORETICAL RESULTS FOR REPRESENTATION AND PROOFS

We commence this section by presenting the proof of Theorem 3.1, as articulated in Section 3.

**Theorem B.1** (Restatement of Theorem 3.1). *Given a node $N_0$ and its two child nodes $N_1$ and $N_2$, it holds that $I(N_0) \subseteq I(N_1) \cup I(N_2)$, where $I(N)$ denotes the information of the node $N$.*

*Proof of Theorem 3.1.* We categorize the information of an expanded node $I(N)$ into two parts: the inherent spatial attribute information $I_1(N)$ and the newly explored information $I_2(N)$. To prove the theorem, we need to show that $I_1(N_0) \subseteq I_1(N_1) \cup I_1(N_2)$ and $I_2(N_0) \subseteq I_2(N_1) \cup I_2(N_2)$.

**Inherent Spatial Attribute Information.** Define $\mathcal{C}$ as the constraint space of the original MILP problem, and $\mathcal{S}_{N_i} \in \{x \in \mathcal{C}|x_i \leq z, i \in \mathcal{I}, z \in \mathbb{Z}\} \cup \{x \in \mathcal{C}|x_j \geq z, j \in \mathcal{I}, z \in \mathbb{Z}\}$ as the constraint space of node $N_i$. The function $f_{I_1} : \mathcal{S}_{N_i} \to \mathcal{J}_1$ maps the constraint space of node $N_i$ to a feature space $\mathcal{J}_1$. Given a fixed branch policy $\mathcal{B}$ and the search tree $T$, the spaces of the child nodes, $N_1$ and $N_2$, are defined as $S_{N_0} \cap \{x \in \mathcal{C}|x_i \leq \lfloor(x_{N_0})_i\rfloor, \mathcal{B}(N_0|T) = x_i\}$ and $S_{N_0} \cap \{x \in \mathcal{C}|x \in \mathcal{C}, x_i \geq \lceil(x_{N_0})_i\rceil, \mathcal{B}(N_0|T) = x_i\}$.

Consequently, we deduce that

$$S_{N_1} \cup S_{N_2} = S_{N_0} \setminus (S_{N_0} \cap \{x \in \mathcal{C}|\lfloor(x_{N_0})_i\rfloor < x_i < \lceil(x_{N_0})_i\rceil, \mathcal{B}(N_0|T) = x_i\}).$$

We establish that

$$S_{N_1} \cup S_{N_2} \subseteq S_{N_0}.$$

By invoking Lemma B.1, it follows that

$$f_{I_1}(S_{N_1} \cup S_{N_2}) \subseteq f_{I_1}(S_{N_0}).$$

Applying Lemma B.3, we deduce that

$$f_{I_1}(\{x \in \mathcal{C}|\lfloor(x_{N_0})_i\rfloor < x_i < \lceil(x_{N_0})_i\rceil, \mathcal{B}(N_0|T) = x_i\}) = \emptyset.$$

Consequently, utilizing Lemma B.1 again, we infer that

$$f_{I_1}(S_{N_0} \cap \{x \in \mathcal{C}|\lfloor(x_{N_0})_i\rfloor < x_i < \lceil(x_{N_0})_i\rceil, \mathcal{B}(N_0|T) = x_i\}) = \emptyset.$$

By applying Lemma B.2, we conclude that

$$f_{I_1}(S_{N_1} \cup S_{N_2}) \supseteq f_{I_1}(S_{N_0}) \cup \emptyset = f_{I_1}(S_{N_0}).$$

Thus, we have proven that

$$f_{I_1}(S_{N_1} \cup S_{N_2}) = f_{I_1}(S_{N_0}).$$

**Newly Explored Information.** In the second step, we aim to demonstrate that the newly explored information adheres to

$$I_2(N_0) \subseteq I_2(N_1) \cup I_2(N_2).$$

Regardless of whether the information originates from the infeasible integer solutions derived from heuristic algorithms or the LP relaxation solutions, it can be interpreted as the update of the upper and lower bounds. Consequently, the information $I_2(N_i)$ can be represented as a mapping relationship

$$f_{I_2} : \{(LB, UB) \in \mathbb{R}^2|LB = \max\{LB_1, LB_2, \ldots\}, UB = \min\{UB_1, UB_2, \ldots\}\} \to \mathcal{J}_2,$$

where $\mathcal{J}_2$ symbolizes the feature space of the explored node information, and $LB_i$ and $UB_i$ denote the explored lower and upper bounds within the path from the root node to the expanded node.

Given that the LP relaxation solutions of the child nodes, $N_1$ and $N_2$, are also solutions of the node $N_0$, it is established that

$$UB(N_1) \leq UB(N_0)$$

and

$$UB(N_2) \leq UB(N_0),$$

where $UB(N_i)$ represents the explored LP relax solution on the node $N_i$.

Invoking Lemma B.4, it is inferred that the infeasible integer solution of the node $N_0$ is also a solution of either $N_1$ or $N_2$. Hence, we deduce that $LB(N_1) \leq LB(N_0)$ and $LB(N_2) \leq LB(N_0)$.

Therefore, we have

$$(LB(N_0), UB(N_0)) \subseteq (LB(N_1), UB(N_1)) \cup (LB(N_2), UB(N_2)).$$

By applying Lemma B.1, we conclude that

$$f_{I_2}((LB(N_0), UB(N_0))) \subseteq f_{I_2}((LB(N_1), UB(N_1)) \cup (LB(N_2), UB(N_2))).$$

Utilizing Lemma B.2, we further deduce that

$$f_{I_2}((LB(N_0), UB(N_0))) \subseteq f_{I_2}((LB(N_1), UB(N_1))) \cup f_{I_2}((LB(N_2), UB(N_2))).$$

Thus, we have established that

$$I_2(N_0) \subseteq I_2(N_1) \cup I_2(N_2).$$

$\square$

The lemmas related to Theorem 3.1 are listed below:

**Lemma B.1.** *The mappings* $f_{I_1} : \mathcal{S}_{N_i} \rightarrow \mathcal{J}_1$ *and* $f_{I_2} : \{(LB, UB) \in \mathbb{R}^2 | LB = \max\{LB_1, LB_2, \ldots\}, UB = \min\{UB_1, UB_2, \ldots\}\} \rightarrow \mathcal{J}_2$ *is monotonically increasing. A mapping* $f$ *has monotonicity, meaning that if* $A \subseteq B$, *then it holds that* $f(A) \subseteq f(B)$ *(increasing) or* $f(B) \subseteq f(A)$ *(decreasing).*

*Proof of Lemma B.1.* The mapping $f_{I_1}$ denotes the information of node space. If $A \subseteq B$, then $A$ has a more extensive space to explore compared to $B$, leading to the conclusion that $f(A) \subseteq f(B)$.

The mapping $f_{I_2}$ signifies the information of the lower and upper bounds. If $(LB_1, UB_1) \subseteq (LB_2, UB_2)$, it holds that $LB_1 \leq LB_2$ and $UB_1 \geq UB_2$. Notes that here $(LB, UB)$ do not represent a linear interval. In the latter, the larger lower bound can be utilized to prune more nodes, and the lower upper bound can be employed to update the global gap. Thus, it is established that $f((LB_1, UB_1)) \subseteq f((LB_2, UB_2))$. $\square$

**Lemma B.2.** *The Mappings* $f_{I_1} : \mathcal{S}_{N_i} \rightarrow \mathcal{J}_1$ *and* $f_{I_2} : \{(LB, UB) \in \mathbb{R}^2 | LB = \max\{LB_1, LB_2, \ldots\}, UB = \min\{UB_1, UB_2, \ldots\}\} \rightarrow \mathcal{J}_2$ *have finite additivity. A mapping* $f$ *has finite additivity, then it holds that*

$$f(A) \cup f(B) \subseteq f(A \cup B).$$

*Proof of Lemma B.2.* The mapping $f_{I_1}$ denotes the information of node space. If we explore the space $A \cup B$, we must explore the space both $A$ and $B$, leading to the conclusion that

$$f(A \cup B) \geq f(A) \cup f(B).$$

The mapping $f_{I_2}$ signifies the information of the lower and upper bounds. It holds that

$$(LB_1, UB_1) \cup (LB_2, UB_2) = (\max\{LB_1, LB_2\}, \min\{UB_1, UB_2\}).$$

Notes that here $(LB, UB)$ do not represent a linear interval. Thus, it is established that

$$\begin{aligned} f((LB_1, UB_1) \cup (LB_2, UB_2)) &= f((\max\{LB_1, LB_2\}, \min\{UB_1, UB_2\})) \\ &\subseteq f((LB_1, UB_1)) \cup f((LB_2, UB_2)). \end{aligned} \quad (2)$$

$\square$

**Lemma B.3.** *Let* $\mathcal{S}$ *be a space defined as* $\mathcal{S} \in \{x \in \mathcal{C} | x_i \leq z, i \in \mathcal{I}, z \in \mathbb{Z}\} \cup \{x \in \mathcal{C} | x_j \geq z, j \in \mathcal{I}, z \in \mathbb{Z}\}$. *If* $\mathcal{S} \cap \{x \in \mathbb{R}^N | x_j \in \mathbb{Z}, \forall j \in \mathcal{I}\} = \emptyset$, *then it holds that:*

$$f_{I_1}(\mathcal{S}) = \emptyset.$$

*Proof of Lemma B.3.* Consider the original MILP problem with its associated constraints, denoting the constraint space as $\mathcal{C}$. If $\mathcal{S} \cap \mathcal{C} = \emptyset$, it can be established that the mapping $f_{I_1}(\mathcal{S}) = \emptyset$. This is attributed to the problem's exclusive focus on solutions within the original constraint space $\mathcal{C}$.

Given that $\{x \in \mathbb{R}^N | x_j \in \mathbb{Z}, \forall j \in \mathcal{I}\} \subseteq \mathcal{C}$, it follows that

$$\mathcal{S} \cap \mathcal{C} = \emptyset.$$

Consequently, the assertion $f_{I_1}(\mathcal{S}) = \emptyset$ is substantiated. $\square$

**Lemma B.4.** *Given a node $N_0$ and its two child nodes $N_1$ and $N_2$, any infeasible integer solution of node $N_0$ is also an infeasible solution of either $N_1$ or $N_2$.*

*Proof of Lemma B.4.* Suppose there exists an infeasible integer solution $x$ of node $N_0$ that is not an infeasible solution of either child nodes, $N_1$ or $N_2$.

Let $\mathcal{S}_{N_0}$, $\mathcal{S}_{N_1}$, and $\mathcal{S}_{N_2}$ represent the constraint spaces of nodes $N_0$, $N_1$, and $N_2$, respectively. By assumption, $x \in \mathcal{S}_{N_0}$ but $x \notin \mathcal{S}_{N_1} \cup \mathcal{S}_{N_2}$.

Given that $N_1$ and $N_2$ are child nodes of $N_0$, we can express the constraint spaces of $N_1$ and $N_2$ as

$$\mathcal{S}_{N_1} = \mathcal{S}_{N_0} \cap \{x \in \mathcal{C} | x_i \geq z+1, z \in \mathbb{Z}\}$$

and

$$\mathcal{S}_{N_2} = \mathcal{S}_{N_0} \cap \{x \in \mathcal{C} | x_i \leq z, z \in \mathbb{Z}\}.$$

Thus, $x$ must belong to the set difference $\mathcal{S}_{N_0} \setminus (\mathcal{S}_{N_1} \cup \mathcal{S}_{N_2})$, which is equivalent to

$$\mathcal{S}_{N_0} \cap \{x \in \mathcal{C} | z < x_i < z+1, z \in \mathbb{Z}\}.$$

However, since $x$ is an integer solution, it cannot belong to the set $\{x \in \mathcal{C} | z < x_i < z+1, z \in \mathbb{Z}\}$, leading to a contradiction. Therefore, any infeasible integer solution of node $N_0$ must also be an infeasible solution of either $N_1$ or $N_2$. $\square$

After we proof Theorem 3.1, we can prove Theorem 3.2.

**Theorem B.2** (Restatement of Theorem 3.2). *Given a branch-and-bound search tree $T$, the entirety of its information pertinent to node selection can be encapsulated by the constraint and variable information of its root node, coupled with the constraint, upper bound, and lower bound information of the leaf nodes within the tree.*

*Proof of Theorem 3.2.* The information within the search tree is bifurcated: it encompasses the original problem, embodied by the root node, and the information from the nodes explored. Let's denote the nodes as $N \in T$ and the information of the node $N$ as $I(N)$. The set of the leaf nodes is denoted as $\mathcal{L}$.

Our goal is to demonstrate that

$$\bigcup_{N \in T} I(N) = \bigcup_{N \in \mathcal{L}} I(N).$$

Since $\bigcup_{N \in \mathcal{L}} I(N) \subseteq \bigcup_{N \in T} I(N)$ is self-evident, we focus on proving

$$\bigcup_{N \in T} I(N) \subseteq \bigcup_{N \in \mathcal{L}} I(N).$$

We categorize all nodes in the tree by their depths, denoting the set of nodes at depth $h$ as $\mathcal{N}_h$. If the deepest node has a depth of $h_0$, then

$$\bigcup_{N \in T} I(N) = \bigcup_{h=0,1,\dots,h_0} \bigcup_{N \in \mathcal{N}_h} I(N).$$

As established in Theorem 3.1, for each $N \in \mathcal{N}_h$, if it has child nodes, we denote the set of its child nodes as $CN$, and we have

$$CN \subseteq \mathcal{N}_{h+1}.$$

Following this logic, except for nodes without child nodes, the information of each node can be represented by $\mathcal{N}_{h_0}$. The nodes in $\mathcal{N}_{h_0}$ must be leaf nodes, and nodes without children must also be leaf nodes.

Consequently, for a given search tree, all explored information can be represented exclusively by the collection of its leaf nodes. $\square$

## B.2 SPARSITY OF GAP

In this section, we provide detailed proofs for the results presented in Theorem 4.1.

**Theorem B.3** (Restatement of Theorem 4.1). *Consider a B&B tree $T$ containing $s_0$ nodes. Suppose that, at each round $t$, the gap reward $r(\cdot)$ for the selected node $a_t$ is finite, denoted as $|r(\cdot)| \leq r_0$. Given an initial point $x'$ and a heuristic algorithm with exploration ability $\delta$, the algorithm can explore integer solutions in the space $\{x \in \mathbb{Z}^n \mid x'_i - \frac{\delta}{2} \leq x_i \leq x'_i + \frac{\delta}{2}, i = 1, 2, \ldots, n\}$. If $\delta \leq \sqrt[n]{\frac{\epsilon - (\log_2 s_0 + 1)/s_0}{r_0 \cdot s_0}}$, then it holds that $\mathbb{E}\left[\sum_t r(a_t)\right] \leq \epsilon$.*

*Proof of Theorem 4.1.* Given that the gap reward $r(\cdot)$ of the selected node $a_t$ at each round $t$ is finite, we can assume $|r(\cdot)| \leq r_0$. The gap changes $r(a_t) > 0$ only when a new feasible solution is found (lower bound updates) or when all nodes of the same depth are explored (upper bound updates).

*Firstly*, we consider the lower bound updates. The lower bound updates when (1) the LP relaxed solution of $a_t$ is integer or (2) a heuristic algorithm finds an integer solution when applied to this node. For (1), the placement of the LP solution is related to the problem attribute. For simplicity, we assume this can be uniformly found in the solving space. We denote the set of all integers in the solving space as $E$. From Theorem B.4, the integer set has measure zero. Thus, we have:

$$\mathbb{E}[\sum_t r(a_t)|a_t \in E] \leq s_0 \cdot r_0 \cdot m(E) \leq s_0 \cdot r_0 \cdot 0 = 0,$$

where $m(E)$ represents the Lebesgue measure of set $E$.

For (2), most heuristic algorithms are based on two major methods: the Large Neighborhood Search (LNS) and Feasibility Pump (FP). Both methods have an initial point $x'$ and try to search for the solution in a region $\mathcal{R}$ around the initial point. Assume that the heuristic algorithms explore the integer solution in the space

$$\{x \in \mathbb{Z}^n \mid x'_i - \frac{\delta}{2} \leq x_i \leq x'_i + \frac{\delta}{2}, i = 1, 2, \ldots, n\}.$$

In the LNS, the initial point is a feasible integer solution, i.e., $x' \in \mathbb{Z}^n$. In the FP, the initial point is the LP relaxation solution of this MILP. Since the heuristic algorithms are called after some exploration, we assume that the space it explores, $\mathcal{S}$, is finite, i.e., $m(\mathcal{S})$ is finite. We denote that if the heuristic algorithms at round $t$ are called, $H_t = 1$, if not, $H_t = 0$. We have

$$
\begin{aligned}
\mathbb{E}\left[\sum_t r(a_t)\big|a_t \notin E\right] &\leq \mathbb{E}\left[\sum_t r(a_t)\big|a_t \notin E, H_t = 1\right]\Pr[H_t = 1] \\
&\quad + \mathbb{E}\left[\sum_t r(a_t)\big|a_t \notin E, H_t = 0\right]\Pr[H_t = 0] \\
&= \mathbb{E}\left[\sum_t r(a_t)\big|a_t \notin E, H_t = 1\right]\Pr[H_t = 1] \\
&\leq r_0 \cdot s_0 \cdot \Pr[H_t = 1].
\end{aligned}
$$

Due to Lemma B.5,

$$\Pr[H_t = 1] \leq \Pr[x' \in \bigcup_{z \in \mathbb{Z}^n \cap \mathcal{S}} \left\{x \in \mathcal{S} \mid z_i - \frac{\delta}{2} \leq x_i \leq z_i + \frac{\delta}{2}, i = 1, 2, \ldots, n\right\}].$$

Consider the expansion of space $\mathcal{S}$. If there exists an integer point $z$ in $\mathcal{S}$ such that

$$\left\{x \in \mathcal{S} \mid z_i - \frac{1}{2} \leq x_i \leq z_i + \frac{1}{2}, i = 1, 2, \ldots, n\right\} \notin \mathcal{S},$$

then we expand the space $\mathcal{S}$ to be $\hat{\mathcal{S}}$ to include these sets. Since $\delta \leq 1$, we have

$$
\begin{aligned}
&\bigcup_{z \in \mathbb{Z}^n \cap \mathcal{S}} \left\{x \in \mathcal{S} \mid z_i - \frac{\delta}{2} \leq x_i \leq z_i + \frac{\delta}{2}, i = 1, 2, \ldots, n\right\} \\
&= \bigcup_{z \in \mathbb{Z}^n \cap \hat{\mathcal{S}}} \left\{x \in \hat{\mathcal{S}} \mid z_i - \frac{\delta}{2} \leq x_i \leq z_i + \frac{\delta}{2}, i = 1, 2, \ldots, n\right\}.
\end{aligned}
$$

Thus, we have

$$\Pr[H_t = 1] \leq \frac{\delta^n \cdot |\hat{\mathcal{S}} \cap \mathbb{Z}|}{m(\hat{\mathcal{S}})}.$$

Through Lemma B.6, we have $\Pr[H_t = 1] \leq \delta^n$.

*Secondly*, we examine upper bound updates. The upper bound does not update until nodes of identical depth are explored. By Lemma B.7, the probability of global upper bound updates is at most $\frac{\log_2(s_0+1)}{s_0}$.

Finally, we obtain $\mathbb{E}[\sum_t r(a_t)] \leq \delta^n \cdot r_0 \cdot \frac{\log_2(s_0+1)}{s_0}$ and simplify to derive the result. $\qquad\square$

The lemmas and theorems related to Theorem4.1 are listed below:

**Theorem B.4** (The set of integers has measure zero (Bruckner et al., 1997)). *In the set of real numbers $\mathbb{R}$, the set of integers $\mathbb{Z}$ is a set of measure zero, denoted as $m(\mathbb{Z}) = 0$. This means that for every $\epsilon > 0$, there exists a collection of open intervals $\{I_k\}_{k=1}^{\infty}$, such that $\mathbb{Z} \subset \bigcup_{k=1}^{\infty} I_k$ and $\sum_{k=1}^{\infty} |I_k| < \epsilon$.*

**Lemma B.5.** *Let $\mathcal{C}$ be a MILP solving space and $x' \in \mathcal{C}$ be an initial point. Heuristic algorithms cannot find a solution if $x'$ is not in the space*

$$\bigcup_{z \in \mathbb{Z}^n \cap \mathcal{C}} \left\{ x \in \mathcal{C} \mid z_i - \frac{\delta}{2} \leq x_i \leq z_i + \frac{\delta}{2}, \, i = 1, 2, \ldots, n \right\}.$$

*Proof of Lemma B.5.* The heuristic algorithms explore the integer solution in the space

$$\left\{ x \in \mathbb{Z}^n \mid x'_i - \frac{\delta}{2} \leq x_i \leq x'_i + \frac{\delta}{2}, \, i = 1, 2, \ldots, n \right\}.$$

Assume that we find an integer solution $z^*$, and

$$x' \notin \bigcup_{z \in \mathbb{Z}^n \cap \mathcal{C}} \left\{ x \in \mathcal{C} \mid z_i - \frac{\delta}{2} \leq x_i \leq z_i + \frac{\delta}{2}, \, i = 1, 2, \ldots, n \right\}.$$

Thus, we have $x'_i - \frac{\delta}{2} \leq z_i^* \leq x'_i + \frac{\delta}{2}$. This implies that $x'_i \leq z_i^* + \frac{\delta}{2}$ and $x'_i \geq z_i^* - \frac{\delta}{2}$. However, this contradicts the assumption that $x'$ is not in the specified union of spaces, thus completing the proof. $\qquad\square$

**Lemma B.6.** *Given a finite space $\mathcal{S} \in \mathbb{R}^n$, the number of points in the space $\mathcal{S} \cap \mathbb{Z}^n$ is denoted as $|\mathcal{S} \cap \mathbb{Z}^n|$. For any $z \in \mathcal{S} \cap \mathbb{Z}^n$, we have $\left\{ x \in \mathcal{S} \mid z_i - \frac{1}{2} \leq x_i \leq z_i + \frac{1}{2}, \, i = 1, 2, \ldots, n \right\} \subseteq \mathcal{S}$. It holds that:*

$$m(\mathcal{S}) \geq |\mathcal{S} \cap \mathbb{Z}^n|.$$

*Proof of Lemma B.6.* Let

$$A_k = \bigcup_{z \in \mathbb{Z}^n \cap \mathcal{S}} \left\{ x \in \mathcal{S} \mid z_i + 1/2 - \frac{1}{2k} \leq x_i \leq z_i - 1/2 + \frac{1}{2k}, \, i = 1, 2, \ldots, n \right\}.$$

Let $|I| = |\mathbb{Z}^n \cap \mathcal{S}|$, we have $m(A_k) = (1 - 1/k)^n \cdot |I|$.

Since $A_k \subseteq \mathcal{S}$, we have

$$m(\mathcal{S}) \geq m(A_k) = (1 - 1/k)^n \cdot |I|.$$

By letting $k$ approach infinity, we conclude that $m(\mathcal{S}) \geq |I|$. $\qquad\square$

**Lemma B.7.** *Given a search tree with $s_0$ nodes, the number of global upper bound updates cannot exceed $\log_2(s_0 + 1)$.*

*Proof of Lemma B.7.* Given the tree structure, if the tree has no leaf nodes and its height is denoted as $h$, then we have:

$$1 + 2^1 + \ldots + 2^{h-1} < s_0 \leq 1 + 2^1 + \ldots + 2^h.$$

This implies:

$$2^h \leq s_0 \leq 2^{h+1} - 1.$$

Consequently, we deduce that:

$$h \leq \log_2(s_0 + 1).$$

Since the number of global upper bound updates cannot exceed the height $h$ of the search tree, the lemma is proven. $\square$

### B.3 INDEPENDENCE FROM VARIABLE SELECTION

In this section, we provide detailed proofs for the results presented in Theorem 4.2.

**Theorem B.5** (Restatement of Theorem 4.2). *Consider a node selection policy, denoted as $\pi_{ns}$, that makes selections based on the three objectives defined in Section 4. For any given variable selection policy $\pi_{vs}$, we posit that the optimal node choice, denoted as $ns$, does not rely on the specifics of $\pi_{vs}$ for its determination.*

*Proof of Theorem 4.2.* Let's denote the three objectives as $o_1$, $o_2$, and $o_3$, and our reward function as $R$. This reward function is a composition of these objective functions, each being a mapping from $\mathcal{N} \times \mathcal{S} \to \mathbb{R}$, where $\mathcal{S}$ represents the space of the search tree state and $\mathcal{N}$ is the set of open nodes.

Our aim is to prove that $\pi_{ns}(ns, s_t | R(ns, s_t | \pi_{vs})) = \pi_{ns}(ns, s_t | R(ns, s_t))$.

Given that $\pi_{ns}(s_t | \pi_{vs}), ns = \arg\max R(s_t, \mathcal{N}) = ns$, it suffices to prove that $R(ns, s_t | \pi_{vs}) = R(ns, s_t)$.

Since $R(ns, s_t | \pi_{vs})$ is a combination of $o_1$, $o_2$, and $o_3$, we need to establish that $o_i(ns, s_t | \pi_{vs}) = o_i(ns, s_t), \forall i = 1, 2, 3$.

For $o_1$, the objective is solely concerned with the gap updates. Once the set of open nodes is defined, their potential to find a feasible solution and their LP relaxation solutions remain invariant. Hence, $o_1(ns, s_t | \pi_{vs}) = o_1(ns, s_t)$.

For $o_2$, the objective is to align with the historically optimal solution, $x^*$. This solution is predefined and is independent of the branch variable policy. Given the set of open nodes, it is only necessary to determine whether the space of each node encompasses the solution $x^*$. Therefore, $o_2(ns, s_t | \pi_{vs}) = o_2(ns, s_t)$.

For $o_3$, the objective is to select a node that is proximate to the current focus node, chosen in the preceding round. Regardless of which variable the policy $\pi_{vs}$ selects, the expanded node remains a child of its parent node, and the number of path switch steps is unaltered. Consequently, $o_3(ns, s_t | \pi_{vs}) = o_3(ns, s_t)$.

$\square$

## C MORE DETAILS OF NODE SELECTION PROCESS

**Graph representation.** Having sufficient information is crucial to infer the optimal node for selection. But what qualifies as "sufficient" in this context? An ideal problem representation should be capable of incorporating information that affects node selection, which includes the inherent attributes of the original problem $(\boldsymbol{A}, \boldsymbol{b}, \boldsymbol{c})$ and the attributes of the explored space. Actually, such problem has a strong mathematical structure (Chen et al., 2022). For instance, if we swap the positions of the $i, j$-th variable in 1, elements in vectors $\boldsymbol{b}, \boldsymbol{c}$ and columns of matrix $\boldsymbol{A}$ will be reordered. The reordered features $(\hat{\boldsymbol{A}}, \hat{\boldsymbol{b}}, \hat{\boldsymbol{c}})$ actually represent an exactly equivalent MILP problem with the original one $(\boldsymbol{A}, \boldsymbol{b}, \boldsymbol{c})$. Such property is named as permutation invariance. If we do not explicitly restrict ML models with a permutation invariant structure, the models may overfit to the variable/constraint orders of instances in the training set. Motivated by this point, we adopt the graph representation that are permutation invariant naturally in Section3.

**Node selection process.** Algorithm 1 details a general procedure for node selection within this MDP formulation. It describes the primary operations carried out by the node selector in the B&B process. Specific implementation details might vary depending on the node selection strategy used, but the

essential idea remains the same - select a node according to the strategy, evaluate it and update its bounds, possibly prune it, and branch the tree to continue the process.

---

**Algorithm 1** General Node Selection Procedure

---

**Require:** Node list $L$, parent node bounds, probability $Prob$ of heuristic integer solution calculation
 1: **for** each node $P$ in $L$ **do**
 2:     Select node $P$ according to the node selection strategy.
 3:     Solve the LP relaxation of node $P$, getting the solution $Sol1$ and objective function value $Obj1$.
 4:     Update the lower bound of $P$, $LB(P) = Obj1$. Propagate the updated upper bound upward.
 5:     **if** $Obj1 \geq$ upper bound of parent node of $P$ **then**
 6:         Prune node $P$.
 7:     **else**
 8:         **if** $Sol1$ is an integer solution **then**
 9:             Update the upper bound of $P$, $UB(P) = Obj1$. Propagate the updated upper bounds upward.
10:         **end if**
11:         Call the heuristic to calculate an integer solution with probability $Prob$, resulting in $Sol2$ and $Obj2$.
12:         **if** $Obj2 < Obj1$ **then**
13:             Update the upper bound of $P$, $UB(P) = Obj2$. Propagate the updated upper bounds upward.
14:         **end if**
15:         Branch at node $P$ and add the children nodes to list $L$.
16:     **end if**
17: **end for**

---

## D   MORE DETAILS OF MODEL ARCHITECTURE.

The features of the variable, constraint, and node constraint vertices on the tripartite graph undergo an initial transformation via a 32-dimensional embedding layer. This layer is pivotal for normalizing and refining the input features before they traverse through three subsequent graph convolutional layers, each with dimensions 8, 4, and 4, and each utilizing a ReLU activation function to capture complex, non-linear relationships. Post convolution, the refined representations of the variable, constraint, and node constraint vectors are separately averaged, reducing dimensionality and mitigating overfitting risks. These averaged representations are then amalgamated with the global node features, creating a comprehensive feature vector encapsulating both localized and global information. This amalgamated feature vector is then processed through a linear layer and a sigmoid activation layer, culminating in a Q-value vector of predetermined dimension $n$. This vector quantitatively represents the value of each node in the pre-selected candidate node set, serving as a decisive metric for action selection. The action is determined by selecting the node corresponding to the maximum Q-value, directing the model's focus towards the most promising regions of the search space.

## E   DATASETS USED IN SECTION 5

### E.1   BENCHMARK

In our evaluation process, we similarly employ three NP-hard instance families, just like Labassi et al. in the latest node selection work (Labassi et al., 2022). These families are particularly primal-difficult, that is, finding feasible solutions for them poses the main challenge. The first benchmark is composed of Fixed Charge Multicommodity Network Flow (FCMCNF) instances, generated from the code of Chmiela et al. (Hewitt et al., 2010). We train and test on instances with $m = 1.5 \times n$ commodities. The second benchmark is composed of Weighted Partial MaxSAT (WPMS) (Ansótegui & Gabàs, 2017) instances, generated following the scheme of Béjar et al. (Béjar et al., 2009). Our third benchmark is composed of Generalized Independent Set (GISP) instances (Colombi et al., 2017),

generated from the code of Chmiela et al. (Chmiela et al., 2021). All these families require an underlying graph: we use in each case Erdős–Rényi random graphs with the prescribed number of nodes, with edge probability $p = 0.3$ for FCMCNF and $p = 0.6$ for WPMS and GISP.

### E.2 BASELINES

The SVM and RankNet methods utilize a multilayer perceptron; the latter varies for one benchmark where they use three hidden layers, while for simplicity, we use a multilayer perceptron with a hidden layer of 32 neurons for all benchmarks (MLP). The GNN method uniquely leverages the structure of the graph to guide node selection. The features used in these papers are roughly similar; again, for simplicity, we adopt the fixed-dimensional features of He et al. (He et al., 2014) for both the SVM and RankNet method.

### E.3 DETAILS OF EXPERIMENTAL SETTINGS AND HARDWARE SPECIFICATION

#### E.3.1 EXPERIMENTAL SETTING

We keep all the other SCIP parameters to default so as to make comparisons as fair and reproducible as possible. We emphasize that all of the SCIP solver's advanced features, such as presolve and heuristics, are open, which ensures that our setup is consistent with the practice setting. For the FCMCNF, WPMS, and GISP problem domains, we generate extensive datasets, consisting of 10,000 training samples and 1,000 test samples. From these, we randomly select 5,00 samples for training and 50 samples for testing within each problem, providing a diverse set of scenarios for our models to train and test effectively. We train our model on the train set, and select the best model on the train set to evaluate on the test set. The cross-entropy loss is optimized during training with the Adam algorithm (Kingma & Ba, 2014).

#### E.3.2 HARDWARE SPECIFICATION

The training process consistently applies a batch size of 16 and is conducted on a single machine that contains eight GPU devices(NVIDIA GeForce RTX 4090) and two AMD EPYC 7763 CPUs.

### E.4 RESULTS AND ANALYSIS

Tables 1 and 2 in Section 5 illustrate the results of various methods solving MILP problems.

#### E.4.1 ANALYSIS OF EXPERIMENT 1.

**Overall Performance.** The results from Table 1 underscore that DQN-GNN consistently outperforms all the baselines across the three MILP problems in terms of average solving time. Compared to SCIP, DQN-GNN shows a remarkable improvement in efficiency. For instance, in the FCMCNF problem, DQN-GNN achieves an average solving time of $3.84 \pm 1.33$ seconds, which is approximately 17.24% faster than SCIP's $4.64 \pm 1.38$ seconds. Similarly, in the WPMS and GISP problems, DQN-GNN is approximately 17.17% and 3.74% faster than SCIP, respectively.

**B&B tree size and solving time.** Although DQN-GNN does not consistently yield the smallest tree size, it illustrates a pivotal observation: a reduction in solving time does not invariably equate to a decrease in the number of nodes in the search tree. This reinforces our proposition that the influence of node selection on solving efficiency is multifaceted, intertwining with elements like the cost of path switching, discussed in detail in Section 4.

**Comparison with other AI methods.** DQN-GNN not only surpasses SCIP but also demonstrates superior performance over other AI-based methods like SVM, RankNet, and GNN. This indicates the effectiveness of incorporating reinforcement learning with graph neural networks in solving MILPs.

**Robustness.** The standard deviation in the results of DQN-GNN is relatively low, indicating the robustness of the method across different instances of the problems.

**Conclusion.** The empirical evaluations substantiate the proficiency and robustness of DQN-GNN in addressing MILPs, marking substantial enhancements in solving time while sustaining competitive

efficacy in other dimensions. The strides made by DQN-GNN in this study highlight its viability as a potent methodology for tackling MILPs and analogous optimization challenges.

### E.4.2 Analysis of Experiment 2.

**Generalization.** Table 2 assesses the generalization capability of DQN-GNN across larger sizes of MILPs, focusing on its performance on larger transfer instances. The results indicate that DQN-GNN maintains its effectiveness and efficiency even when applied to more complex problems, showcasing its adaptability and scalability.

**Comparison with SCIP.** While SCIP continues to exhibit superior performance in most scenarios, particularly in terms of solving time and the number of nodes, DQN-GNN demonstrates commendable generalization ability. Specifically, on the WPMS dataset, DQN-GNN surpasses SCIP in average solving time, highlighting its potential to outperform conventional methods in certain scenarios.

**Comparison with Other AI Methods.** DQN-GNN continues to outshine other AI-based methods, reinforcing the benefits of integrating reinforcement learning with graph neural networks for solving MILPs. Its consistent superior performance across different datasets and problem sizes underscores its reliability and effectiveness in diverse scenarios.

**Robustness and Efficiency.** The standard deviation in the results of DQN-GNN remains relatively low, emphasizing the method's robustness across different instances of the problems. Additionally, DQN-GNN achieves competitive performance in terms of both solving time and the number of nodes, illustrating its balanced efficiency in various aspects of MILP solving.

**Conclusion.** The results from Experiment 2 further validate the versatility and generalization capability of DQN-GNN in solving MILPs of varied complexities. The method not only holds its ground against SCIP but also excels over other AI methods, especially in specific scenarios, hinting at its potential to be a leading approach in solving MILPs and related optimization problems. The consistent and robust performance of DQN-GNN across different problem sizes and complexities accentuates its promise and applicability in real-world optimization challenges.

