# OpenReview forum: "Learning Node Selection via Tripartite Graph Representation in Mixed Integer Linear Programming"
_ICLR.cc/2024/Conference — Submitted to ICLR 2024_

### Official Review · Reviewer_pDE5 · 2023-10-30

**Soundness:** 3 good
**Presentation:** 3 good
**Contribution:** 2 fair
**Rating:** 3
**Confidence:** 4

**Summary:**

This paper proposed a method to learn node selection policy for solving mixed integer linear programming (MILP) problems. It tries to resolve two issues in existing deep learning based MILP methods, i.e., insufficient state representation and reward design. For the first issue, a tripartite graph representation scheme is proposed, which differs from the widely used bipartite graph representation with a new set of additional constraints. For the seconde issue, a new reward scheme with three components are designed to facilitate reinforcement training. Theoretical analysis are provided to support the design. Experimental results show that the learned policy outperforms default SCIP policy and several deep learning based methods.

**Strengths:**

1. The motivation is clear, and the two technical contributions are based on interesting and insightful observations.
2. Some theoretical results are presented.
3. The paper is well-written and easy to follow.

**Weaknesses:**

1. The evaluation is relatively weak. The problem instances are too simple, which can be solved by SCIP within a minute. Using a heavy deep learning architecture to enhance MIP solving performance on such trivial problems are not very meaningful. Besides, the speedup effects over default SCIP is not very significant. If the decision maker can allow one minute for solving, then 10 seconds shorter runtime is not very meaningful.

2. The trained policy degrades significantly on larger problem, which further lowers the pracicability of the proposed method.

3. There is no ablation study results to support the advantages of the proposed method. For example, how about using the traditional bipartite graph representation instead of the newly proposed tripartite scheme? What are the contributions of the three reward componenets on the final performance?

**Questions:**

Please see the above weaknesses.

---

> ### Author Response · Authors · 2023-11-22
> **Response to reviewer pDE5**
>
> Dear Reviewer,
>
> Thank you very much for your insightful comments and constructive suggestions. Your feedback is invaluable in enhancing the quality of our paper.
>
> For weaknesses:
>
> We acknowledge your concerns about the simplicity of the problem instances used in our experiments and the relative significance of the speedup achieved. While it is true that the instances we selected can be solved by SCIP within a minute, and our improvements in absolute terms may not seem substantial, we believe there is still value in these results: It is important to recognize that even small gains in efficiency can be impactful in scenarios where decisions are iteratively made over numerous instances. In such contexts, the cumulative time savings achieved through even marginally faster solutions can be significant [1]. For instance, in operational settings where decisions need to be made quickly and repeatedly, a reduction of a few seconds per instance can lead to considerable time savings over thousands of iterations.
>
> Thank you for your insightful comments on the evaluation aspects of our paper. We recognize the importance of conducting extensive experiments and ablation studies to validate the effectiveness of our proposed method. We are actively working on supplementing our experiments to address these gaps. However, due to time constraints, we were not able to complete these additional experiments in time for this submission. We are committed to continuing this work and will prioritize completing these evaluations in the near future.
>
> Reference
>
> [1] Xavier Á S, Qiu F, Ahmed S. Learning to solve large-scale security-constrained unit commitment problems[J]. INFORMS Journal on Computing, 2021, 33(2): 739-756.

---

### Official Review · Reviewer_gXzc · 2023-10-30

**Soundness:** 1 poor
**Presentation:** 2 fair
**Contribution:** 2 fair
**Rating:** 3
**Confidence:** 4

**Summary:**

This paper studies the use of machine learning to improve node selection in branch-and-bound solvers for integer programming. Node selection (along with variable selection and cut selection) is one of the important aspects of branch-and-bound, and a good node selection policy is critical to reducing solve time and the size of the branch-and-bound tree. The authors make two main contributions: first, they propose a new representation of the MIP as a tripartite graph that captures the branching constraints used to arrive at a particular child node; second, they train an RL model based on this representation to tune their node selection policy.

**Strengths:**

The integration of machine learning with MIP solvers is a topic of considerable importance and recent interest. Node selection is an important aspect of this that has not been as thoroughly studied. The authors provide good motivation for studying the problem. Furthermore, consideration of path switching costs in node selection is a subtle and important issue, and it is interesting that the authors study methods to alleviate this.

**Weaknesses:**

The tripartite graph representation in Section 3.2 could be explained a lot more clearly. As is, it is just defined as an abstract object but the authors never define clearly how it is built from the underlying MIP. It is more or less possible to infer how they do this from Figure 1, but all of this ought to be clearly defined in the body of the paper.

Similarly, the information of a node I(N) is never formally defined.

I’m not really sure what Theorem 3.2 is saying. Branch-and-bound is a deterministic algorithm, so indeed all the information needed for its execution is contained at the root node for a fixed node selection policy. A brief glance at the proofs in the appendix makes me skeptical that Theorems 3.1 and 3.2 are saying anything rigorous/meaningful about branch-and-bound.

I do not understand Theorem 4.2 and its proof. The authors claim here that node selection and variable selection are “independent”. The proof seems to show that given a fixed tree state, the optimal node does not depend on the variable selection policy. But the assumption that there is a given fixed tree state means that the variable selection decisions have already been made at the point of node selection. Varying the variable selection policy would change the current subtree, and the premise of the theorem wouldn’t hold. I don’t think it is scientifically valid to claim from this theorem that node selection and variable selection are “independent”, and my guess is that most viewpoints in MIP solving would take issue to this claim.

The writing in the paper needs a lot of improvement. There are grammatical errors and issues with sentence structure in nearly every paragraph that make the paper difficult to read.

**Questions:**

I don’t really understand the motivation in the “From Nodes to Constraints” paragraph. Variable selection is what determines the actual additional constraints to the subproblem; node selection is about exploration order.

Doesn’t the set NC in the tripartite graph representation grow very large since every leaf node is stored? Are there any limitations due to this?

Additional comments/questions:
-Section 2 title “Preliminary” needs to be “Preliminaries” or “Preliminary background” or something like that

-“feature vector of vertex e_k” -> “feature vector of edge e_k” on page 3.

-In the section “MILP as Weighted Bipartite Graph” you might want to include that there is an edge between a variable vertex and a constraint vertex iff the variable has a nonzero coefficient in the constraint.

-The P(X, f) notation in the section “Branch-and-Bound Method” seems out of place. Why not stick with the MIP notation used previously?

-In Section 3.2 “V ∪ C = V ∪ NC = C ∪ NC = ∅” the unions should be intersections.

-In Section 3.2 you should include how the tripartite graph is actually defined, i.e. how are the edges in the graph determined. As far as I can tell the reader is left to infer this from Fig. 1.

-The feature vector labels in Fig. 1 are not explained anywhere as far as I could tell. E.g., what does a feature vector of $(0, +\infty, -2)$ for the node corresponding to variable $x$ mean? Also, the edges between $V$ and $NC$ are labeled with pairs denoting the sign of the branching constraint, which I take to be the edge feature vectors. But these are labeled $e_i^{NC}$ in Fig. 1, which are the edge weights? What are the actual edge weights, and what do they represent?

---

> ### Author Response · Authors · 2023-11-22
> **Response to reviewer gXzc (1/3)**
>
> Dear Reviewer,
>
> Thank you very much for your insightful comments and constructive suggestions. Your feedback is invaluable in enhancing the quality of our paper.
>
> For weakness 1:
>
> We appreciate your constructive feedback on Section 3.2 regarding the tripartite graph representation. In response to your comments, we have revised the manuscript to include a detailed explanation of how the tripartite graph is built step-by-step.
>
> For weakness 2:
>
> Thanks for your insightful comments. In the revised version, we have comprehensively added a more detailed explanation of the definition, rationale, and utility of node information in Section 3.1.
>
> For weakness 3:
>
> Thank you for your insightful observation regarding Theorem 3.2. It appears there may be a misunderstanding regarding the focus and implications of the theorem, which I would like to clarify.
>
> Your comment rightly points out that branch-and-bound (B&B) is a deterministic algorithm and, with a fixed node selection policy, the outcome is determined by the information at the root node. However, the scenario we address in our theorem is different. Theorem 3.2 asserts that, regardless of the node selection strategy employed at any specific round of the B&B algorithm, all information obtained through the exploration of nodes up to that point can be effectively represented using only the information from the root node and the current leaf nodes. It means that for the purpose of representing the state of the B&B tree at any given round, it is not necessary to record detailed information about all the intermediate nodes that have been explored. Instead, it suffices to focus on the root node and the leaf nodes, which carry the most recent branching constraints and updated bounds. This theorem significantly streamlines the representation of the B&B process. By focusing on the root and leaf nodes, we avoid the redundancy of recording extensive information about every intermediate node, thereby reducing computational overhead and improving efficiency.

---

> > ### Author Response · Authors · 2023-11-22
> > **Response to reviewer gXzc (2/3)**
> >
> > For weakness 4:
> >
> > Thank you for your insightful comments and for pointing out the need for clarity regarding Theorem 4.2. Your expertise in this area is greatly appreciated.
> >
> > We apologize for any confusion caused by our use of the term "independent" in our initial draft. In the revised version of the paper, we have adjusted our terminology to more accurately reflect the relationship between node selection and variable selection within the branch-and-bound (B&B) process.
> >
> > Regarding Theorem 4.2, our intention was to highlight that while the variable selection process shapes the tree, the node selection strategy can independently assess the tree's current state to determine the next course of action. As you correctly point out, variable selection decisions indeed influence the formation of the current subtree, and varying variable selection policies would result in different subtrees.
> >
> > The core idea we wish to convey is that once a specific subtree is formed, the node selection process focuses solely on this existing state of the tree to identify the most promising node for exploration. This decision-making process within the node selection phase does not require direct knowledge of the variable selection policies that led to the current state of the tree. It is this aspect of the process that we sought to characterize as being 'independent' of the variable selection strategy.
> > The contribution of this theorem lies in addressing a common concern in prior research: the interdependence of node and variable selection in B&B algorithms. While it is true that variable selection influences the shaping of the subtree and, consequently, the options available for node selection, our theorem clarifies that the node selection strategy itself does not need to factor in the specifics of the variable selection decisions. Instead, it can effectively operate based on the current state of the tree, regardless of the variable selection path that led to it.
> >
> > For weakness 5:
> >
> > Thank you for pointing out the writing issues. We have thoroughly reviewed and revised the paper to enhance its readability and clarity.

---

> > > ### Author Response · Authors · 2023-11-22
> > > **Response to reviewer gXzc (3/3)**
> > >
> > > For question 1:
> > >
> > > Thank you for your insightful comment regarding the motivation behind our approach in the "From Nodes to Constraints" paragraph. We apologize for any confusion caused by our previous explanation and appreciate the opportunity to clarify our intent.
> > >
> > > In our research, we are particularly focused on refining the representation of newly explored information in MILP problems within the branch-and-bound (B&B) process. Our primary objective is to determine which information should be retained and accurately represented at each stage of the search tree. The conventional bipartite graph model (Labassi et al., 2022) comprehensively represents each node as an independent subproblem, but it often results in significant informational redundancy. The overlap in constraints and variables among nodes leads to repetitive representations, which can conceal the subtle yet essential differences between nodes arising from specific branching decisions.  Our design of the tripartite graph retains only the root node's bipartite graph and other leaf nodes' added branching constraints. This approach aims to represent each node's information more accurately and to save a substantial amount of problem information typically redundantly stored in bipartite graphs.
> > >
> > > For question 2:
> > >
> > > Thank you for your astute observation regarding the potential size of the set NC in our tripartite graph representation. Indeed, as the number of leaf nodes in the branch-and-bound tree increases, the corresponding set of node constraints (NC) could potentially grow large.
> > >
> > > However, it's important to note that we only need to store a limited amount of information for each leaf node—specifically, the unique constraints added due to branching and the upper and lower bounds associated with these nodes. This quantity of information, while seemingly extensive, does not significantly exceed the storage requirements compared to previous approaches where each node is treated as an independent MILP problem.
> > >
> > > Moreover, it is crucial to understand that these leaf nodes represent the current options available for node selection in the branch-and-bound process. Prior to making a selection, it is indeed necessary to represent the characteristics of each option comprehensively. Therefore, despite the potential size of NC, representing this information is essential for effective decision-making within our proposed framework.
> > >
> > > Additionally, we are actively exploring methods to further reduce and optimize the storage of this information without compromising the effectiveness of node selection in our framework.
> > >
> > > For other question：
> > >
> > > I would like to express my gratitude for pointing out the issues related to the clarity and expression in my paper. Accordingly, I have made necessary revisions to address these concerns, striving for clearer and more precise language. Your constructive comments have significantly contributed to enhancing the quality of this paper.

---

### Official Review · Reviewer_ghtC · 2023-11-02

**Soundness:** 3 good
**Presentation:** 3 good
**Contribution:** 3 good
**Rating:** 6
**Confidence:** 3

**Summary:**

The authors introduce a novel tripartite graph representation for the branch-and-bound search tree, which is theoretically proven to encapsulate sufficient information of the search tree, enabling effective node selection in the solving process. They propose three innovative metrics for node selection and develop a DQN-GNN model that employs reinforcement learning to learn node selection policies in MILP solving. The model consists of a GNN that encodes the tripartite graph representation of the search tree into a fixed-dimensional vector and a DQN that takes the encoded vector as input and outputs a Q-value vector for each candidate node. The model is trained to maximize the expected cumulative reward, which is defined as a combination of three metrics: the node's contribution to the relaxation bound improvement, the node's contribution to the search tree size reduction, and the node's contribution to the solution quality improvement. The authors design and conduct experiments that demonstrate the efficacy of the DQN-GNN model in enhancing the efficiency of solving MILPs, showing significant improvement over existing human-designed and learning-based benchmarks.

**Strengths:**

1. Introducing a novel tripartite graph representation for the branch-and-bound search tree, which is theoretically proven to encapsulate sufficient information of the search tree, enabling effective node selection in the solving process.
2. Proposing three metrics for evaluating rewards of node selection and developing a DQN-GNN model that employs reinforcement learning to learn node selection policies in MILP solving.
3. The model exhibits commendable generalization to larger MILPs.

**Weaknesses:**

1.	Objective 2: The historical optimal solution path is not clear. I don't quite understand what is a historical optimal solution. Is it a global optimal solution or just an intermediate non-proven local optimal solution?
2.	Since the performance is inconsistent on larger cases, can author provide any insights on what type of MILP problem would be suitable for the proposed methods?
3.	What is the rule of setting hyperparameter n (# of nodes for selection)? What if I choose some other values (e.g., n=30,50,100)
4.	Do you also restrict the number of nodes to be selected for SCIP?

**Questions:**

Please refer to Strengths and Weaknesses.

---

> ### Author Response · Authors · 2023-11-22
> **Response to reviewer ghtC (1/2)**
>
> Dear Reviewer,
>
> Thank you very much for your insightful comments and constructive suggestions. Your feedback is invaluable in enhancing the quality of our paper.
>
> For weakness 1 :
>
> This is great comment! In previous versions, we did not provide enough details about related works.
> First, let me briefly outline the existing works:
> Most current machine learning studies in node selection rely on initially using a solver to solve a set of training instances, obtaining optimal solutions for each instance. These methods treat each node in the branch and bound tree as a subproblem and determine whether the pre-solved optimal solution lies within that subproblem's constraint space, thus assessing if the node is on the path to an optimal solution. Their main strategy involves imitation learning, where nodes on the path to the optimal solution are deemed worthy of selection, while others are considered inferior. These methods primarily rely on manually designed features or treating each node as a subproblem represented by a bipartite graph.
> Now, let me explain how our work differs in two significant aspects:
> (1) (Section 3) Previous works compile each tree node into a bipartite graph, in which most of the information is redundant. We aim for AI to learn the distinctions between these nodes, such as the minor constraint changes caused by different branching. We adopt a tripartite graph representation, which offers a more detailed and nuanced portrayal of each node within the branch and bound tree. Our design of the tripartite graph retains only the root node's bipartite graph and other leaf nodes' added branching constraints. This approach aims to represent each node's information more accurately and to save a substantial amount of problem information typically redundantly stored in bipartite graphs.
> (2) (Section 4) In designing the reward function, we refer to previous works' reliance on historical optimal solutions (Our Objective2), which provides valuable and explicit information aiding AI methods in learning to find an optimal solution. However, these methods often overlook the possibility of multiple optimal solutions, where a node not leading to optimal solution 'a' might lead to optimal solution 'b'. Furthermore, these approaches typically do not consider finding suboptimal solutions to narrow the gap (Our Objective1), which could accelerate the convergence of the solving process. They also fail to account for the cost of switching paths between nodes in the search for a more likely route to the optimal solution (Our Objective3). Our work uniquely addresses these aspects, providing a more holistic approach to node selection that is not solely dependent on tracing a pre-determined path to a single optimal solution.
>
> For weakness 2:
>
> Thank you for your insightful question, which has prompted valuable reflection on our part regarding the performance of our methods across different types of MILP problems.
> Our approach is designed to be generally applicable to a broad range of MILP problems. However, as you rightly pointed out, we observe varying performance depending on the specific characteristics of each problem. We appreciate the inspiration your question has provided and, it reflects a common phenomenon where a general-purpose model exhibits varying performances across different tasks. This variation could be linked to deeper issues of interpretability within GNN models. While a definitive answer might be complex, we can, based on our experience and observations, offer some conjectural insights.
>
> We hypothesize that our method may be more suitable for MILP problems where the relationship between decision variables and constraints is more straightforward and less obscured by complex interrelations. An example of where our method might show more promise is in problems like the Weighted Partial MaxSAT (WPMS). In such problems, the structure may allow for clearer constraints, which our method could potentially leverage more effectively.

---

> > ### Author Response · Authors · 2023-11-22
> > **Response to reviewer ghtC (2/2)**
> >
> > For weakness 3:
> >
> > Thank you for raising this excellent question. The primary reason for setting a parameter $n$ is to manage the dynamic variation in the number of leaf nodes within the branch and bound tree. By imposing a limit on $n$, we constrain the action space of our reinforcement learning model to a manageable subset of nodes, facilitating more efficient learning and decision-making processes. In our experiments, we opted for a relatively large value of $n$, ensuring that it was generally greater than or equal to the actual number of leaf nodes. Our observations suggest that when $n$ is adjusted around the number of leaf nodes, the results remain largely consistent, indicating a degree of robustness in this approach.
> >
> > However, it is important to acknowledge that we did not extensively explore scenarios where $n$ is significantly smaller. The impact of a reduced $n$ on our results might depend on the heuristics used for pre-selecting nodes to be included in this subset.
> >
> > For weakness 4:
> >
> > Thank you for your question！In our approach, we do not impose a restriction on the number of nodes to be selected specifically for SCIP. The rationale behind this decision is that our method for pre-selecting the top $n$ nodes, based on their scores, leverages SCIP's default node selection strategies. In such a setup, if we were to pre-select the top $n$ nodes using the same rules and then choose the highest-scoring node from this subset, the outcome would essentially be identical to directly selecting the top node according to SCIP's default process. This is because the node that emerges as the highest-scoring in both scenarios is the same. Therefore, introducing a step to limit the selection to $n$ nodes within SCIP would be redundant.

---

### Official Review · Reviewer_cMLb · 2023-11-03

**Soundness:** 2 fair
**Presentation:** 2 fair
**Contribution:** 2 fair
**Rating:** 3
**Confidence:** 3

**Summary:**

The paper addresses the heuristic designs for node selection in BnB for MILP from an ML-guided perspective. The paper proposes a new tripartite representation and a new RL-based framework to learn such heuristics. It provides theoretical analyses to support some of the design choices in the framework. Empirically, the method is tested on three different benchmark, resulting in shorter runtime and smaller tree sizes in some cases.

**Strengths:**

1. The paper studies the important questions of node selection for MIP solving.
2. It provides a theoretical understanding that motivates the design of the ML methods
3. Experiments show promising results on small MIP instances.

**Weaknesses:**

1. The empirical study was done on pretty easy instances and the slowest approaches took only less than half a minute to solve them. It is more common for this kind of study to test on harder and/or larger instances.

2. Only runtime and number of nodes were reported in evaluations. It will be important to include more metrics such as win/loss rate and scatter plots to understand the per-instance performance.

3. The paper doesn’t provide a literature review of related work. There are three very related works (SVM, Ranknet and GNN) compared against in experiment. However I wish to understand what the unique contributions of this paper are over those previous works. Furthermore, there are a few more works on ML-guided node selection for tree search algorithms for other combinatorial optimization problems. It would be important to acknowledge them in related work.

4. Writing can be improved. (1) The edges between V and NC are never defined. I think they are defined similarly to the edges between V and C; (2) Gap reward in Theorem 4.1 is undefined (though defined later); (3) It would be important to describe the intuitive idea for proving each theorem. (4)Typo:
	Section 3.2: V \cup C = V\cup NC = C\cup NC should be taking the intersection instead.
	Section 4.2 mordern -> modern

**Questions:**

The node constraints vertices represent constraints added in addition to the root problem. They seem to be only one variable if they are constraints added by branching, as mentioned in 3.2 (the 4th line in the paragraph). In that case, each NC vertex will have only one edge connecting to one variable vertex. Is the tripartite representation necessary? This would be more useful if you consider cuts added to the nodes.

---

> ### Author Response · Authors · 2023-11-22
> **Response to reviewer cMLb (1/2)**
>
> Dear Reviewer,
>
> Thank you very much for your insightful comments and constructive suggestions. Your feedback is invaluable in enhancing the quality of our paper.
>
> For weakness 1 and 2:
>
> We sincerely appreciate the valuable comment made by the reviewers that have helped enhance our paper. We acknowledge that our initial experiments were conducted on relatively easier instances, primarily due to time and computational resource constraints. We are currently in the process of expanding our experiments to include more challenging datasets, specifically following the approaches used in He et al. (2014) and Nair et al. (2020). The medium datasets being incorporated include MIK (Atamtürk, 2003), which consists of a set of MILP problems with knapsack constraints, and CORLAT (Gomes et al., 2008), a real dataset utilized for constructing a wildlife corridor for grizzly bears in the Northern Rockies region. This will include not only runtime and the number of nodes but also more detailed metrics such as win/loss rates.
> Due to time constraints, these additional experiments are still underway and have not been completed for inclusion in this revision. However, we are committed to providing a more comprehensive evaluation in our final version, should the paper be accepted.
>
> For Weakness3:
>
> This is great comment! In previous versions, we did not provide enough details about related works.
> First, let me briefly outline the existing works:
> Most current machine learning studies in node selection rely on initially using a solver to solve a set of training instances, obtaining optimal solutions for each instance. These methods treat each node in the branch and bound tree as a subproblem and determine whether the pre-solved optimal solution lies within that subproblem's constraint space, thus assessing if the node is on the path to an optimal solution. Their main strategy involves imitation learning, where nodes on the path to the optimal solution are deemed worthy of selection, while others are considered inferior. These methods primarily rely on manually designed features or treating each node as a subproblem represented by a bipartite graph.
> Now, let me explain how our work differs in two significant aspects:
> (1) (Section 3) Previous works compile each tree node into a bipartite graph, in which most of the information is redundant. We aim for AI to learn the distinctions between these nodes, such as the minor constraint changes caused by different branching. We adopt a tripartite graph representation, which offers a more detailed and nuanced portrayal of each node within the branch and bound tree. Our design of the tripartite graph retains only the root node's bipartite graph and other leaf nodes' added branching constraints. This approach aims to represent each node's information more accurately and to save a substantial amount of problem information typically redundantly stored in bipartite graphs.
> (2) (Section 4) In designing the reward function, we refer to previous works' reliance on historical optimal solutions (Our Objective2 in Section 4.1), which provides valuable and explicit information aiding AI methods in learning to find an optimal solution. However, these methods often overlook the possibility of multiple optimal solutions, where a node not leading to optimal solution 'a' might lead to optimal solution 'b'. Furthermore, these approaches typically do not consider finding suboptimal solutions to narrow the gap (Our Objective1 in Section 4.1), which could accelerate the convergence of the solving process. They also fail to account for the cost of switching paths between nodes in the search for a more likely route to the optimal solution (Our Objective3 in Section 4.1). Our work uniquely addresses these aspects, providing a more holistic approach to node selection that is not solely dependent on tracing a pre-determined path to a single optimal solution.
>
> For weakness 4:
>
> Thank you for pointing this out. We made corrections and clarifications in the revision.
>
> For Question 1:
>
> Specifically, concerning the necessity of the tripartite representation: although each leaf node indeed adds only one constraint relative to its parent node due to branching, these constraints, relative to the original problem, accumulate over several rounds of branching. Therefore, representing these subproblems (leaf nodes) in contrast to the original problem's constraint space requires multiple edges connecting to variable nodes, not just one.
>
> Once again, we are grateful for your valuable feedback. We look forward to further guidance to improve our work.

---

> ### Author Response · Authors · 2023-11-22
> **Response to reviewer cMLb (2/2)**
>
> Reference
>
> [1] He H, Daume III H, Eisner J M. Learning to search in branch and bound algorithms[J]. Advances in neural information processing systems, 2014, 27.
>
> [2] Nair V, Bartunov S, Gimeno F, et al. Solving mixed integer programs using neural networks[J]. arXiv preprint arXiv:2012.13349, 2020.
>
> [3] Atamtürk A. On the facets of the mixed–integer knapsack polyhedron[J]. Mathematical Programming, 2003, 98(1-3): 145-175.
>
> [4] Gomes C P, Van Hoeve W J, Sabharwal A. Connections in networks: A hybrid approach[C]//Integration of AI and OR Techniques in Constraint Programming for Combinatorial Optimization Problems: 5th International Conference, CPAIOR 2008 Paris, France, May 20-23, 2008 Proceedings 5. Springer Berlin Heidelberg, 2008: 303-307.

---

> > ### Comment · Reviewer_cMLb · 2023-11-23
> >
> > I have read the rebuttal. Since a couple of my main concerns (weaknesses 1-2) haven't been addressed, my evaluation remains the same.

---

### Meta-Review · Area_Chair_WsyN · 2023-12-04

**Metareview:**

There were major concerns with this paper. Particularly, this is am empirical paper and it does not give sufficient evidence for the problem method for solving IPs.

**Justification For Why Not Higher Score:**

The paper has issues with writing and, further, there was a lack of results to suggest the method proposed is interesting over prior work.

**Justification For Why Not Lower Score:**

N/A

---

### Decision · Program_Chairs · 2024-01-16

Reject